



# Drivers of the variability of the isotopic composition of water vapor in the surface boundary layer

Jelka Braden-Behrens[1], Lukas Siebicke[1], and Alexander Knohl[1,2]

[1]University of Goettingen, Bioclimatology, Faculty of Forest Sciences and Forest Ecology, Germany
[2]University of Goettingen, Centre of Biodiversity and Sustainable Land Use (CBL), Germany

**Correspondence:** Jelka Braden-Behrens (jbraden1@gwdg.de)

**Abstract.**

Measurements of the isotopic composition of water vapor, $\delta_v$, as well as measurements of the isotopic composition of evaporation and transpiration provide valuable insights in the hydrological cycle. Here we present measurements of $\delta_v$ in the surface boundary layer (SBL) in combination with eddy covariance (EC) measurements of the isotopic composition of

evapotranspiration $\delta_{ET}$ for both $\delta D$ as well as $\delta^{18}O$ over a full growing season above a managed beech forest in central Germany. Based on direct measurements of isoforcing IF and the height $h$ of the planetary boundary layer (PBL), we provide an estimate of isoforcing-related changes in $\delta_v$, revealing the influence of local evapotranspiration (ET) on $\delta_v$. At seasonal time scales we find no evidence for a dominant control of $\delta_v$ by local ET. Rayleigh distillation could at most explain 35 % of the observed variability and we did not find indications for the influence of entrainment at seasonal time scales. Instead, we

obtain a strong significant correlation ($R^2 \approx 0.52$; $p < 10^{-35}$) of $\delta_v$ to temperature. We conclude that the observed seasonal variability of $\delta_v$ is neither dominated by Rayleigh processes, entrainment nor local ET but likely linked to other temperature-related processes such as fractionation during evaporation. At a diurnal time scale we find that even during summer, when transpiration is high and at a height of only 10 m above the canopy, ET is overruled by entrainment effects throughout the day from approximately 10 am to 4 pm. ET only dominates the diurnal cycle of $\delta_v$ in the mornings and evenings. Thus, from

diurnal to seasonal time scale, ET does not dominate the measured $\delta_v$ at our field site, even if the measurements were carried out close to the canopy. We further conclude, that accounting for PBL height $h$ is essential to understand drivers of $\delta_v$.

## 1 Introduction

The isotopic composition of water vapor ($\delta_v$) in the atmosphere can provide insights into the hydrological cycle at scales ranging from the leaf scale to the global scale (Gat, 1996; Jouzel et al., 2000; Yakir and Sternberg, 2000; Wen et al., 2010;

Huang and Wen, 2014). Potential drivers of the temporal variability of $\delta_v$ are the origin and the different histories of air masses (Noone et al., 2013), the removal of rain from the atmosphere, (see e.g. Gat, 2000) as well as horizontal advection and land-sea breeze circulation (Lee et al., 2006). Further, $\delta_v$ is driven by transpiration and evaporation (see e.g. Gat, 2000) and vertical atmospheric mixing, including the growth and decay of the planetary boundary layer (PBL) (Lee et al., 2012a) and in particular entrainment (e.g. Lee et al., 2007).





At seasonal time scales the removal of precipitation from the atmosphere is a major driver of $\delta_v$. This is a complex process that influences $\delta_v$ via the origin of air masses (Ambach et al., 1968), the thermodynamic conditions during cooling (see e.g. Dansgaard, 1964), fractionation during condensation, isotopic exchange between rain drops and the surrounding air and evaporation from rain drops (see e.g. Dansgaard, 1964) as well as the sublimation of snow (Noone et al., 2013). Empirically, these processes yield the 'amount effect', i.e. small amounts of rain are enriched, and the 'temperature effect', i.e. higher

condensation temperature is related to higher $\delta$-values of precipitation, (see e.g. Dansgaard, 1964). In a simplified model the removal of rain from the atmosphere can be described as a Rayleigh distillation process (see e.g. Gat, 2000; Lee et al., 2006), i.e. the removal of material from an open system (Gat, 1996). Rayleigh distillation yields a log-linear relationship between the isotopic composition of water vapor $\delta_v$ and its mole fraction $C_{H2O}$ (see e.g. Lee et al., 2006) [1].

$$\delta_v = \delta_{v,0} + (\alpha - 1) \times \log\left(\frac{C_{H2O}}{C_{H2O,0}}\right) \qquad (1)$$

With the (temperature dependent) equilibrium fractionation factor $\alpha > 1$ and the subscript 0 as a label for the values of source vapor (before Rayleigh rain-out). Such a log-linear relationship has been found by many authors with $R^2$ values (see table 2) ranging from approximately 0.15 (Huang and Wen, 2014) to 0.78 (Lee et al., 2006; Wen et al., 2010) indicating different importance of Rayleigh processes as drivers of the seasonal variability of $\delta_v$.

At diurnal time scales local ET and/or entrainment dominate the diurnal variability of $\delta_v$ in the SBL at different sites (see

e.g. Lai and Ehleringer, 2011; Huang and Wen, 2014; Noone et al., 2013). Entrainment is a mixing process driven by the gradients at the capping inversion between the PBL and the free atmosphere (Lee et al., 2012a). Thus, entrainment transports dry and isotopically depleted air into the PBL (Lee et al., 2012a). Transpiration on the other hand, the major part of ET at the global scale (Jasechko et al., 2013), isotopically enriches[2] water vapor in the SBL (Lee et al., 2006). Further, transpiration and evaporation involve various (temperature dependent) fractionation processes of ET (Gat, 2000), and selective ET from different

water sources (Gat, 2000; McDonnell, 2014).

Measurements of $\delta_v$ have been carried out at different heights above various ecosystems (see e.g. Huang and Wen, 2014, for a list of such measurement sites), but only few studies discuss the importance of ET and entrainment based on directly measured $\delta_{ET}$ (and/or the corresponding isoforcing), see table 2 for an overview. These studies obtain diurnal cycles that are dominated by entrainment (Lee et al., 2007; Griffis et al., 2010; Lai and Ehleringer, 2011), ET (Zhang et al., 2011; Huang and

Wen, 2014), both (Lee et al., 2012a; Welp et al., 2012; Noone et al., 2013), or both in combination with dew formation (Welp et al., 2008). At seasonal time scales, different importance of Rayleigh processes as drivers of the seasonal variability of $\delta_v$ were obtained: The $R^2$ value of the corresponding log-linear relationship ranges from approximately 0.15 (Huang and Wen, 2014) to 0.78 (Lee et al., 2006; Wen et al., 2010). Evidence of the importance of ET in the PBL water budget was found based on modeling and flux gradient measurements of $\delta^{18}O_{ET}$ and $\delta D_{ET}$ (Huang and Wen, 2014; Griffis et al., 2016). In particular,

---

[1] This refers to the linearized form of the Rayleigh equation.

[2] This statement is mentioned by Lee et al. (2006) particularly for $\delta^{18}O$. For $\delta D$, it can be derived from the above cited statement for $\delta^{18}O$ in combination with the fact that transpiration is not expected to change local deuterium excess $d_{local}$ (Gat, 1996).




even at a height of 185 m above a cropgrassland canopy, Griffis et al. (2016) estimate the relative contribution of ET to range from 0 to close to 100 %, with a median of 34% (Griffis et al., 2016).

Forest ecosystems have comparably large ET, thus, their contribution to the local variability of $\delta_v$ is expected to be more pronounced than for other ecosystems. Direct measurements of $\delta^{18}O_v$ and $\delta D_v$ in general and in particular above forest ecosystems are sparse. Only five of the studies in Table 2 (Lee et al., 2007; Welp et al., 2008; Griffis et al., 2010; Huang
and Wen, 2014; Griffis et al., 2016) are based on measurements of $\delta_{ET}$ and direct eddy covariance (EC) measurements of $\delta^{18}O_{ET}$ have been carried out only above corn/soybean canopy (Griffis et al., 2010, 2011). A forest ecosystem is only captured by one of these studies (Huang and Wen, 2014), based on flux-gradient measurements. Given the limitations of flux gradient

**Table 1.** Nomenclature and abbreviations

| | Stable isotope specific notations |
|---|---|
| $R_{std}$ | Isotopic ratio of a standard material |
| $\delta$ | $(R_{meas} - R_{std})/R_{std}$ |
| $\delta_v$ | Isotopic composition of water vapor |
| IF | Isoforcing; IF := $\overline{w'\delta'}$ |
| | Abbreviations |
| PBL | Planetary boundary layer |
| SBL | Surface boundary layer |
| EC | Eddy covariance |
| ET | Evapotranspiration |
| WVIA | Water vapor isotope analyzer |
| 2 Hz-HF-WVIA | WVIA for 2 Hz measurements (with high flow plumbing) |
| WVISS | Water vapor isotope standard source |
| HF-WVISS | WVISS optimized for high flow |
| DPG | Dew point generator |
| VSMOW | Vienna standard mean ocean water |
| IRMS | Isotope ratio mass spectrometry |
| GMWL | Global meteoric water line |
| LMWL | Local meteoric water line |
| | Meteorological quantities |
| RH | Relative humidity |
| VPD | Vapor pressure deficit |
| TKE | Turbulent kinetic energy |
| $u^*$ | Friction velocity |
| $C_{H20}$ | Water vapor mole fraction |
| $h$ | Height of the PBL |



**Table 2.** Overview of studies that discuss diurnal and seasonal dynamics of $\delta_v$ based on different methods: flux gradient(FG), eddy covariance (EC), repeated profile measurements (RPM), time series analysys (TS) or large eddy simulations (LES). These studies discuss entrainment (ENT), evapotranspiration (ET), dew formation (dew) and land-sea breeze as majos driver of the diurnal variability of $\delta_v$. The $R^2$ value of the log linear relationship between $\delta^{18}O_v$ varies from 0.15 to 0.78. For a list of additional time series measurements of $\delta_v$, see also (Huang and Wen, 2014).

| Author | Measured quantity | Method | Ecosystem | Major driver of $\delta_v$ (diurnal) | $R^2$ value (Rayleigh) | Highest inlet [m] |
|---|---|---|---|---|---|---|
| Lee et al. (2007) | $\delta^{18}O_{ET}$ | FG | tempreate forest | ENT | 0.68 | 30.7 |
| Welp et al. (2008) | $\delta^{18}O_{ET}$ | FG | soy | ET & ENT & dew | 0.66 | 2.0 |
| Huang and Wen (2014) | $\delta^{18}O_{ET},\delta D_{ET}$ | FG | arid oasis | ET | 0.15 | 5.5 |
| Griffis et al. (2010, 2011) | $\delta^{18}O_{ET}$ | EC | corn/soybean | ENT | 0.49 | 3.5 |
| Griffis et al. (2016) | $\delta^{18}O_{ET},(\delta D_{ET})$ | model & FG | crop/grassland | EV & ENT | 0.76 | 185 |
| Lee et al. (2012a) | $\delta^{18}O_v$ (modelled) | LES | mixed | ET & ENT | assumed/modelled | 500 |
| Lai and Ehleringer (2011) | $\delta^{18}O_v\ \delta D_v$ | RPM | forest | ENT | not discussed | 60 |
| Noone et al. (2013) | $\delta^{18}O_v$ | RPM | mixed | ET & ENT | not discussed | 300 |
| Zhang et al. (2011) | $\delta^{18}O_v$ | TS | urban& wheat | ET | 0.7 & 0.5 | 10 &4.2 |
| Welp et al. (2012) | $\delta^{18}O_v$ | TS | forest, grassland urban & agricultural | ET & ENT | not discussed | 1.6 -37 |
| Lee et al. (2006) | $\delta^{18}O_v$ | TS | coastal& forest | land sea breeze, (&ENT) | 0.78 | rooftop |
| Wen et al. (2010) | $\delta^{18}O_v\ ,\delta D_v$ | TS | urban | ENT | 0.78 | 10 |



measurements above forest ecosystems (see e.g. Griffis, 2013), a discussion of the variability of $\delta_v$ in combination with EC measurements of $\delta_{ET}$ over forest ecosystems could provide new insights into potential drivers of $\delta_v$ for both, the diurnal and the seasonal scale.

Here we present time series of $\delta^{18}O_v$ and $\delta D_v$ in the SBL above a managed beech forest in central Germany in combination with direct eddy covariance measurements of ET, $\delta^{18}O_{ET}$ and $\delta D_{ET}$ and standard meteorological measurements. Our measurements were carried out at a height of approximately 10 m above the top of the canopy in the SBL and we expect high transpiration during summer. Our objectives are to evaluate the influence of entrainment and local ET on $\delta_v$ in the SBL close to the canopy of a forest ecosystem. We hypothesize, that at our measurement position, local ET is an important driver of $\delta_v$ at both, the seasonal and the diurnal time scale. As many authors find indications for a pronounced influence of entrainment in various ecosystems (cf. table 2), we further hypothesize, that entrainment and local ET alternately dominate $\delta_v$ at a diurnal time scale.

## 2 Material and methods

### 2.1 Field site

The field site of this study is located in central Germany (51°19'41,58" N; 10°22'04,08" E; approximately 450 meters above sea level) and vegetated by a managed beech forest (dominated by *Fagus sylvatica L.*). The forest in the surroundings of the tower has a relatively homogeneous top-weighted canopy structure (Anthoni et al., 2004; Braden-Behrens et al., 2017). The forest height (defined as the average height of the highest 20 % of the trees) was 37 m in 2004 and the maximum leaf area index was approximately 4 m$^2$m$^{-2}$ (Anthoni et al., 2004).

### 2.2 Stable isotope measurements of $\delta_{H2O}$

We used a customized version of a commercially available water vapor isotope analyzer (2 Hz-HF-WVIA, *Los Gatos Research. Inc.,San Jose, USA*) to measure the water vapor mole fraction $C_{H2O}$ and its stable isotopic compositions $\delta^{18}O_v$ and $\delta D_v$ at 44 m height above the forest floor. The measurement frequency is 2 Hz (Braden-Behrens et al., 2019). The used analyzer is an off-axis integrated-cavity output spectrometer in the near infrared, i.e. a laser-based absorption spectrometer that uses a high-finesse optical cavity to enhance the effective path length (see Aemisegger et al., 2012; Los Gatos Research Inc., 2013, for details on the non-customized instrument). The customization of this analyzer enables us to take measurements with a frequency of 2 Hz at a flow rate of approximately 4.2 slpm (see Braden-Behrens et al., 2019, for details). The 2 Hz-HF-WVIA was calibrated hourly with a customized version of the water vapor isotope standard source for high flow rates (HF-WVISS, *Los Gatos Research. Inc.*,San Jose, USA). In brief, this calibration unit dries and compresses ambient air and mixes it with nebulized liquid water, that is further diluted to yield water vapor with different mole fractions at a constant isotopic composition (see e.g. Aemisegger et al., 2012; Los Gatos Research Inc., 2012). We customized this calibration unit to enable calibration at the analyzer's flow rate of 4.2 slpm over a broad mole fraction range (see Braden-Behrens et al., 2019).



### 2.3 Eddy covariance measurements of $\delta_{ET}$

95 We measured the three dimensional wind velocity and the sonic temperature at 44 m height above the forest floor with a measurement frequency of 20 Hz using a sonic anemometer (Gill-R3, *Gill Instruments*, Lymington, UK). The inlet of the $\delta^{18}O_v$ and $\delta D_v$ measurements was in the vicinity of this anemometer with a 5 cm northward, a 10 cm eastward and a 20 cm vertical separation. We combined the 20 Hz anemometer measurements with the 2 Hz measurements of $\delta^{18}O_v$ and $\delta D_v$ to calculate the magnitude and the isotopic composition of ET using the eddy covariance software EddyPro®, version 6.2.0 (LiCor 100 Biosciences, 2016). Further specification of this setup such as tubing material, tube heating, flow rates and the different data processing settings are described by Braden-Behrens et al. (2019). In brief, the tubing to the 2 Hz-HF-WVIA was made of PTFE (teflon) and heated to avoid condensation. The flow rates in the main tubes were chosen to guaranty turbulent conditions within the tubes upstream of the analyzer. Data processing includes double rotation and block averaging, spike removal, correction for instrument separation, as well as spectral corrections in the high frequency and low frequency range (see Braden-Behrens 105 et al., 2019). Additionally to the stable isotope eddy covariance measurements, the meteorological tower is also equipped with a standard closed path $CO_2$ and $H_2O_v$ analyzer (LI-6262 *LiCor Inc.*, Lincoln, USA). The inlet of this analyzer is located -5 cm northwards, 10 cm eastwards and 20 cm vertically separated from the anemometer. We use this analyzer in combination with the sonic anemometer for standard eddy covariance measurements and to evaluate our stable isotope setup. A direct comparison of both EC setups shows that the two fluxes correlate strongly with each other ($slope \approx 0.8$; $R^2 = 0.9$) (Braden-Behrens et al., 110 2019), but the 2 Hz-HF-WVIA underestimates the net $H_2O_v$ flux by approximately 20 % (Braden-Behrens et al., 2019).

### 2.4 Calculation of isoforcing

We quantify potential effects of ET on the isotopic composition of the atmosphere, by calculating isoforcing, based on EC measurements of the magnitude and the isotopic composition of ET (described in Braden-Behrens et al., 2019). Isoforcing (see e.g. Lee et al., 2009) can be interpreted as the rate of change of the atmospheric $\delta$ value multiplied by the boundary layer 115 height $h$, if a simple isotopic mass balance model (see e.g. Lai et al., 2006) is assumed with only one flux component from the surface and no horizontal advection or entrainment from above (see also Sturm et al., 2012; Braden-Behrens et al., 2019).

$$\text{IF} = \frac{F}{C_a \rho_a}(\delta_F - \delta_v) = h\frac{\mathrm{d}\delta_v}{\mathrm{d}t} \qquad (2)$$

With the flux $F$ (e.g. ET), its isotopic composition $\delta_F$ (e.g. $\delta_{\text{ET}}$), the atmospheric mole fraction $C_a$, the molar density of atmospheric air $\rho_a$, the atmosphere's isotopic composition $\delta_v$ and the height $h$ of the planetary boundary layer (PBL).

### 120 2.5 Additional meteorological and isotopic measurements

The meteorological tower at our site is equipped with standard meteorological measurements (see Anthoni et al., 2004, for details). Among other variables, air and soil temperatures are measured at 2 m and 44 m height and at 2, 8, 16, 32 and 64 cm depth, respectively. Relative humidity (RH), vapor pressure deficit (VPD), precipitation amount, as well as wind direction and velocity are measured at 44 m height. A set of different radiation sensors measure diffuse, up- and downwelling longwave,





shortwave, photosynthetically active radiation and net radiation (see also Braden-Behrens et al., 2019). In this experiment, we
also measured the isotopic composition of rain above the canopy using three integrated rain samplers. These rain samplers are
self-manufactured according to the description of Gröning et al. (2012). Every second week, we took two samples out of each of
the three integrated rain samplers and analyzed them for their $\delta^{18}$O and $\delta$D composition at the centre for stable isotope research
and analysis (*KOSI*, University of Goettingen) using isotope ratio mass spectrometry (IRMS). After the subsamples were taken,
the bottles of the integrated rain samplers were replaced by dry bottles. The uncertainty of our rain sample measurements, as
quantified by the median of the standard deviations of the 6 respective subsamples of each data point was approximately 0.2 ‰
for $\delta^{18}$O and 1.5 ‰ for $\delta$D.

### 2.6 Determination of the PBL height

The time series of the estimate of the height $h$ of the planetary boundary layer (PBL) used in this study is based on a combination
of advanced modeling and data assimilation systems delivered by the ERA5 reanalysis product of the European Centre for
Medium-Range Weather Forecasts, ECMWF. The height of the PBL was obtained for the full year 2016 at the ERA5 product's
native hourly resolution.

The height $h$ of the PBL was extracted from the ERA5 data set at 51.25° latitude, 10.25° longitude, at a distance of 11.94 km
from the study site. The grid point used for uncertainty estimation was located at 51.50° latitude, 10.50° longitude, at a distance
of 21.18 km from the study site. The spatial grid and the temporal resolution of the reanalysis ensemble uncertainty estimate
is coarser (3 h and 0.5 degree resolution) compared to the reanalysis data (1 h and 0.25 degree resolution). Therefore the grid
points of the two products do not necessarily coincide, resulting in a different reference grid point for the estimates of PBL-
height and uncertainty of PBL-height, respectively, for the current study. In both cases we used the grid point closest to the
experimental site.

The following details regarding the PBL data source and processing apply to the current analysis. Here we use the ERA5
reanalysis data set generated by 4D-Var data assimilation in CY41R2 of ECMWF's Integrated Forecast System (IFS). The
components of IFS are described in detail in ECMWF (2016a,c,d,e,f,g,b). In summary, "ERA5 provides hourly estimates of
a large number of atmospheric, land and oceanic climate variables. The data cover the Earth on a 30 km grid and resolve the
atmosphere using 137 levels from the surface up to a height of 80km. ERA5 includes information about uncertainties for all
variables at reduced spatial and temporal resolutions. [...] ERA5 combines vast amounts of historical observations into global
estimates using advanced modeling and data assimilation systems."[3]. Regarding the uncertainty, we present "the uncertainty
as defined for ERA5 by the Ensemble of Data Assimilation (EDA) system [...]. The EDA takes into account mostly random
uncertainties in the observations, sea surface temperature (SST) and the physical parametrization of the model [...] systematic
model errors are not taken into account by the EDA and the errors (uncertainties) as defined by the EDA are uncorrelated.
Furthermore, for affordability reasons the EDA has a lower resolution than ERA5 itself, so the EDA system is unable to

---

[3]https://www.ecmwf.int/en/forecasts/datasets/reanalysis-datasets/era5



directly describe all the uncertainties of ERA5. [...] Nevertheless, comparison of uncertainties provides excellent information on when and where the reanalysis products are more or less accurate [...]" [4].

## 3 Results and discussion

### 3.1 Diurnal variability of $\delta_v$ and potential drivers

In spring and summer the obtained diurnal cycles of $\delta D_v$, $\delta^{18}O_v$ and $C_{H2O}$ resemble a sine curve, while in autumn, there are only shallow and rather noisy diurnal cycles (Fig. 1). In summer, when the diurnal cycle is most pronounced, $\delta_v$ is enriched from midnight to approximately 10 am (GMT+1)[5] followed by a depletion throughout the day until 4 pm by approximately 1‰ for $\delta^{18}O$ and 4.5‰ for $\delta D$.The diurnal cycle of isoforcing values (calculated with Eq. 2) is dominated by the diurnal cycle of ET with a concave shape that is more pronounced in spring and summer and less pronounced in autumn (Fig. 1).

Further, all obtained isoforcing values are positive, thus local ET corresponds to an enrichment of $\delta_v$. In summer, the period with the most pronounced diurnal cycles, the amplitude of isoforcing values is approximately 0.08 and 0.6 m‰ s$^{-1}$ for $\delta^{18}O$ and $\delta D$ respectively (Figure 1). Comparable magnitudes of isoforcing values were found by other authors (Welp et al., 2008; Lee et al., 2007; Hu et al., 2014) in different ecosystems including a temperate forest and a semi-arid area. Diurnal cycles of $\delta_v$ that are similar to the observed sine-shaped diurnal cycle have been obtained at field sites with less transpiration such

as an urban cite in Bejing (Zhang et al., 2011), or an artificial oasis cropland in the late growing season (Huang and Wen, 2014). However, a similar diurnal cycle can also be found in a large eddy simulation (Lee et al., 2012a,b). Diurnal cycles of $\delta_v$ over forest ecosystems have been sparsely measured. However, they have been reported to show constant, concave or convex shapes (Lee et al., 2006, 2007; Welp et al., 2012) and did not always agree between $\delta^{18}O_v$ and $\delta D_v$ (Welp et al., 2012; Lai and Ehleringer, 2011).

The isoforcing-related change in ambient water vapor $d\delta_v/dt_{iso}$ can be estimated by dividing isoforcing (IF) by the PBL height $h$ (Eq. 2). This estimation is based on assuming a well mixed PBL with constant $\delta_v$.[6] Because both, IF and $h$ approach zero at nighttime, the estimated $d\delta_v/dt_{iso}$ shows large peaks at nighttime. These artifacts are related to dividing by zero. Thus the estimation of $d\delta_v/dt_{iso}$ only provides meaningful values at daytime. Throughout the day, from 8 am to 5 pm, the estimated $d\delta_v/dt_{iso}$ fluctuates around 0.1 permil for $\delta^{18}O$ and 1.1 permil for $\delta D$, with slightly lower values in autumn (see Fig. 2).

The directly measured diurnal cycles of $d\delta_v/dt_{meas}$ do not agree with the isoforcing-related estimate $d\delta_v/dt_{iso}$ (see Fig. 2). In particular in spring and summer, we measure negative values of $d\delta_v/dt$ around midday, associated with a depletion of ambient water vapor, while the isoforcing-related change $\delta_v$ always yields an enrichment. However, the magnitude of the diurnal cycle of $d\delta_v/dt_{meas}$ is comparable to the isoforcing-related estimate $d\delta_v/dt_{iso}$.

The mean diurnal cycles of our dataset (Fig. 1 and Fig. 2) show conclusively that isoforcing due to local ET does not dom-

inate the diurnal variability of $\delta D_v$ and $\delta^{18}O_v$. This is also the case during summer, when transpiration is high - even if our

---

[4]https://confluence.ecmwf.int/pages/viewpage.action?pageId=111158117

[5]All times given in this paper refer to local winter time (GMT+1).

[6]This assumption is probably not justified, but the calculation of $d\delta_v/dt$ should provide a rough estimate of the influence of local ET on $\delta_v$.





**Figure 1.** Mean diurnal cycles of water vapor mole fraction $C_{H2O}$ and its isotopic composition $\delta_v$, in combination with isoforcing IF and evapotranspiration ET for spring (March to May), summer (June to August) and autumn (September to November).







**Figure 2.** Mean diurnal cycles of the measured (black) and estimated (red) change in $\delta_v$ in combination with diurnal cycles of PBL height $h$ and turbulent kinetic energy TKE for spring (March to May), summer (June to August) and autumn (September to November).

measurements took place only 10 m above the tree tops. Thus, the measured data do not support the hypothesis that during summer, local ET dominates the diurnal variability of $\delta_v$ over a forest ecosystem. We further conclude that due to the large variability of the boundary layer height $h$, it is essential to account for $h$ when estimating the influence of local ET on ambient water vapor. A discussion of the influence of local ET that is purely based on isoforcing IF overlooks the influence of boundary

layer mixing processes. Further, the concurrent trends in the diurnal cycles of $C_{H2O}$ and $\delta_v$ (Fig. 1) indicate, that entrainment dominantly influences $\delta_v$ from the forenoon to the afternoon: The rise in $\delta_v$ in the morning and in the afternoon is related to a water source (e.g. local ET) whereas the decrease of $\delta_v$ throughout the day is related to mixing with dryer and isotopically lighter air (entrainment). Despite our measurement position beeing closely above the forest canopy and despite high transpiration, we observe this indication for a dominant influence of entrainment from the forenoon to the afternoon also in summer.





The diurnal cycles of PBL height $h$, TKE and $C_{\text{H2O}}$ (Fig. 1 and Fig. 2) further indicate that the diurnal cycle of $\delta_v$ is alter-
nately dominated by entrainment and local ET. As turbulence increases boundary layer mixing, we suppose that larger TKE
corresponds to larger entrainment. This is also supported by the strong agreement between the diurnal cycles of TKE and PBL
height $h$ (Fig. 2). They co-vary throughout the day with an increase in the mornings, when the PBL grows, and a maximum
around 2 pm. The observed negative values for $d\delta_v/dt_{iso}$ throughout the day occur during a time period with large TKE (Fig.
2). Additionally, the amplitude of TKE is largest in summer, medium large in spring and comparably small in autumn (Fig. 2).

### 3.2  Seasonal variability of $\delta_v$ and potential drivers

Over the growing season, the hourly averaged water vapor mole fraction $C_{\text{H2O}}$ at 44 m height varied from approximately 4000
to 25 000 ppm, while the corresponding isotopic compositions $\delta_v$ varied from approximately -33 to -12‰ for $\delta^{18}O_v$ and from
$-147$ to -12‰ for $\delta D_v$ (Fig. 3). Similar ranges for $\delta^{18}O_v$ and $\delta D_v$ over a full year have been measured at different field
sites (e.g. Welp et al., 2012; Huang and Wen, 2014; Griffis et al., 2016). Potential processes that could drive the observed
seasonal variability of $\delta_v$ are local ET, rain-out (Rayleigh distillation), selective water use by plants and temperature dependent
fractionation. Here we test for the importance of different drivers at the seasonal scale by analyzing linear regressions betreen
$\delta_v$ and quantities that are related to these processes (c.f. Table 3.

At our measurement position in the SBL (at 44 m height, i.e. only 7 m above the approximately 37 m high forest) local ET
could be an important driver of $\delta_v$ because the measurements are carried out close to the evaporating source. To evaluate if
local ET drives the seasonal variability of $\delta_v$, we calculated the correlations between daily averaged $\delta_v$ and the respective local
isoforcing IF as well as the midday value[7] of isoforcing-related change in ambient water vapor $d\delta_v/dt_{iso}$ (Fig. 5). Over the
whole measurement period (period: 'all times' in Fig. 5) the calculated correlations between $\delta_v$ and isoforcing (IF $^{18}$O and
IF D) are insignificant (p>0.1). However, for the time period between leaf unfolding and leaf senescence in fall (period: 'green
leaves' in Fig: 5), when we expect transpiration, there are significant ($p < 10^{-5}$) but weak ($R^2 \approx 0.17$) correlations between
$\delta_v$ and the corresponding isoforcing values IF as well as the isoforcing related change $d\delta_v/dt_{iso}$ ($R^2 \approx 0.2$) for both, $\delta D_v$
and $\delta^{18}O_v$. We propose that these weak correlations between isoforcing-related quantities do not reflect a causal relationship.
Additionally to being weak, they are all negative (as shown in Fig. 5). If local ET drove the isotopic composition of ambient
water vapor, we would obtain a positive correlation between IF and $\delta_v$. Thus, we discard the hypothesis that at the field site of
our measurement campaign ET dominates the seasonal variability of $\delta_v$. We further conclude, that it is important to use direct
measurements of IF and an estimation of $d\delta_v/dt_{iso}$ to not over-interpret correlations between $\delta_v$ and $\delta_{\text{ET}}$. In our dataset, we
indeed find a positive moderate ($R^2 \approx 0.4$) and significant ($p < 10^{-25}$) correlation between $\delta D_{\text{ET}}$ and $\delta D_v$, but not for $\delta^{18}O_{\text{ET}}$
and $\delta^{18}O_v$ (see Table 3).

Rayleigh rain-out is another potentially important driver of the seasonal variability of $\delta_v$. Rayleigh distillation processes
yield a log-linear relationship between $\delta_v$ and $C_{\text{H2O}}$ (Eq. 1) as tested in Fig. 6. Throughout the whole measurement period,
both $\delta_v$ values correlate only moderately ($R^2 \approx 0.35$), but significantly ($p < 10^{-25}$) to $\log(C_{\text{H2O}})$. This correlation is weaker

---

[7]Here we choose the midday value of $d\delta_v/dt_{iso}dt$ to reflect $\int d\delta_v/dt_{iso}dt$. The reason for this choice is that due to data gaps and due to the peaks in
$d\delta_v/dt_{iso}$ at nighttime, the numeric integration of $d\delta_v/dt_{iso}$ would not be meaningful.





**Figure 3.** Time series of the measured water vapor mole fraction $C_{H2O}$ (black) and its isotopic composition in $\delta^{18}O$ (dark blue) and $\delta D$ (light blue) in combination with rain above the canopy (cyan), temperature in 2 m height (red), relative humidity (light grey) and PBL height $h$ (dark gray). The vertical lines mark the times of the beginning of leaf unfolding on 19. April 2016 and the beginning of leaf senescence on 6. October 2016.

in summer (see Table 3). Stronger correlations between $\delta_v$ and $\log(C_{H2O})$ at the seasonal time scale have been reported for various other sites, for $\delta^{18}O$ (Lee et al., 2006, 2007; Welp et al., 2008; Wen et al., 2010; Zhang et al., 2011; Griffis et al., 2016) and also for $\delta D$ (Wen et al., 2010; Zhang et al., 2011). Similarly small correlation coefficients ($R^2 < 0.17$) that get weaker







**Figure 4.** Time series of evapotranspiration ET (orange) and the isoforcing related to ET in $\delta^{18}O$ (dark gray) and $\delta D$ (light gray) in combination with the corresponding change in atmospheric delta values $d\delta_v/dt$ for $\delta^{18}O$ (dark green) and $\delta D$ (light green). The vertical lines mark the times of the beginning of leaf unfolding on 19. April 2016 and the beginning of leaf senescence on 6. October 2016.

during summer, were obtained above an arid artificial oasis (Huang and Wen, 2014). At our field site, the correlation between $\delta_v$ and $\log(C_{\text{H2O}})$ is dominated by the periods before leaf unfolding and after leaf senescence (Fig. 6). During the period between leaf unfolding and senescence, this correlation is particularly weak, with $R^2$ values below 0.2 (Table 3). Thus, our

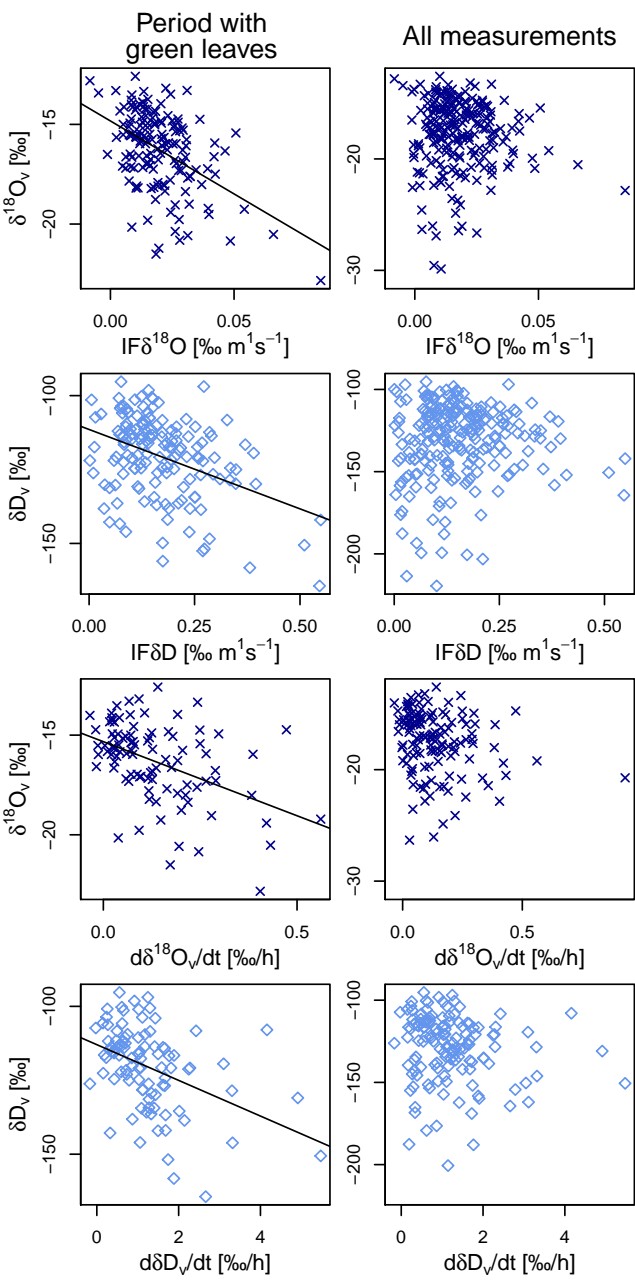

**Figure 5.** The isotopic composition of vater vapor $\delta_v$ plotted against isoforcing IF on a diurnal time scale. The black lines are significant linear regressions with $R^2 \approx 0.25$, $p < 10^{-7}$ for $\delta D$ and $R^2 \approx 0.33$, $p < 10^{-10}$ for $\delta^{18}O$.

measurements imply that at our field site with large transpiration in summer, Rayleigh distillation might drive some variability in $\delta_v$, but in particular during summer Rayleigh distillation does not dominate changes in $\delta_v$.


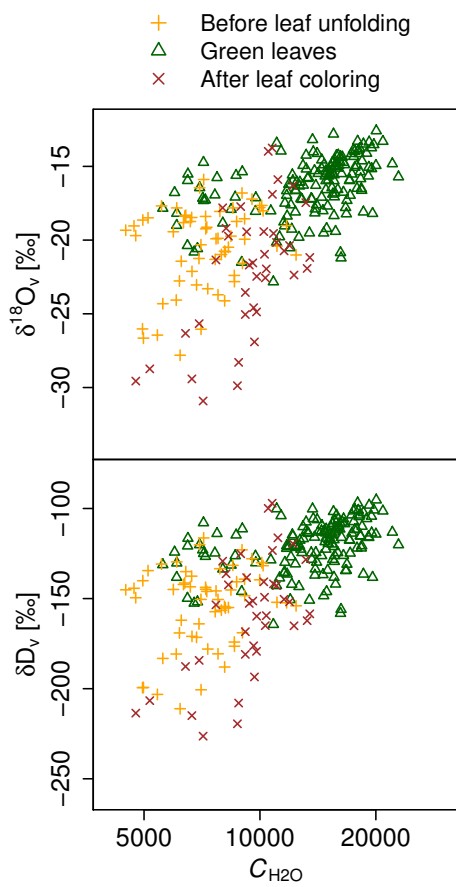

**Figure 6.** Semi-logarithmic plot of $\delta_v$ against $C_{H2O}$, based on diurnal averages. The different symbols are different periods: The period before leaf unfolding (orange crosses), the period with green unfolded leaves (green triangles) and the period after the beginning of leaf senescence in fall (brown diagonal crosses). A log-linear relationship would indicate a system that is dominated by Rayleigh distillation.

We plot the measured $\delta_v$ values in the $\delta^{18}O$-$\delta D$-plane (Fig. 7), to evaluate deviations from Rayleigh distillation processes. Therefore, we analyze deviations of measured $\delta_{rain}$ and $\delta_v$ from the global meteoric water line[8] (GMWL, $\delta D = 8\delta^{18}O + 10$) (Craig, 1961; Dansgaard, 1964; Gat, 2000; Bowling et al., 2017), that corresponds to equilibrium fractionation during Rayleigh rain-out (Dansgaard, 1964). At our field site, the the local meteoric water line (LMWL), as defined by the linear regression of rain sample data (Dansgaard, 1964; Bowling et al., 2017), has a slope of approximately $7.4 \pm 0.3$. This lower slope of the

LMWL compared to the GMWL might reflect the influence of non-Rayleigh-distillation processes such as local evaporation (from open water bodies) and selective transpiration but also evaporation from falling raindrops (Gat, 2000). However, the LMWL is to be interpreted cautiously, as the underlying rain samples span different seasons (see Gat, 1996). Here we use the

---

[8]The GMWL is based on meteoric waters, i.e. precipitation and waters from rivers and lakes (except East African rivers and lakes) and waters from closed basins (Craig, 1961).



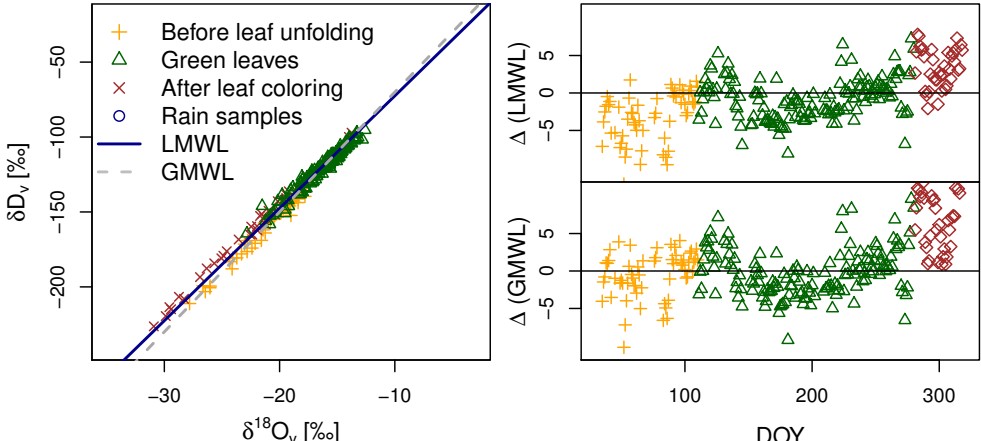

**Figure 7.** Plot of the diurnal averages of the isotopic composition of water vapor in the $\delta^{18}$O-$\delta$D-plane in combination with the GMWL and the LMWL (left panel). The right panels show the deviations $\Delta$ from the GMWL and LMWL plotted against the day of the year (DOY). Different symbols represent different periods: The period before leaf unfolding (orange crosses), the period with green unfolded leaves (green triangles) and the period after the beginning of leaf senescence in fall (brown diagonal crosses). Before leaf unfolding $\delta_v$ fluctuates around the GMWL and after leaf unfolding $\delta_v$ gets closer to the LMWL.

LMWL and the GMWL only to compare the slopes of the measured isotopic composition of water vapor with these lines (Fig. 7). The dual isotope analysis reveals that the measured $\delta_v$ values over the season formed the clusters: Before leaf unfolding

on DOY 110, the measured (daily averaged) $\delta_v$ values followed the GMWL (with a slope of $8.0 \pm 0.2$), whereas after leaf senescence on DOY 280, the slope in the $\delta^{18}$O-$\delta$D-plane was $7.3 \pm 0.1$ and thus closer to the LMWL. For the period with green leaves, an even lower slope of $6.9 \pm 0.1$ was measured. This becomes more visible when deviations from the GMWL and LMWL are plotted versus time (Fig. 7, right panels). While the GMWL with its slope of 8 seems to describe the variability of $\delta_v$ before leave unfolding, the obtained values after leaf senescence in fall are on average better represented by the LMWL,

indicating some influence of local conditions or fluxes. Between leaf unfolding and leaf senescence in fall, the variation in the measured difference from both the LMWL and the GMWL shows a seasonal cycle which might be related to seasonal shifts in the source (but in general also in the fractionation) of water vapor.

To reveal other processes that drive the seasonal variability of $\delta_v$, we calculate the Pearson correlation coefficient $R_{\text{pear}}$ for different meteorological quantities (Table 3). We find no indication that the seasonal variability of $\delta_v$ is driven by the variability

of entrainment, as we find no significant correlation between $\delta_v$ and turbulent kinetic energy TKE and friction velocity $u^*$ (see Table 3). For the whole measurement period, the observed seasonal variability of $\delta_v$ was strongly correlated to temperature ($R^2 > 0.5$, $p < 10^{-35}$). This correlation was stronger than the correlation to $\log(C_{\text{H2O}})$, discussed above as an indicator for Rayleigh distillation processes. In particular, the observed correlation with temperature at 2 m height above the ground had an $R^2$ of 0.52 and was slightly stronger than the correlation with temperature at 44 m above the ground for both isotopic species.

Similarly, soil temperature at 2 cm depth is much stronger correlated to $\delta_v$ than soil temperature at 64 cm depth. These height





**Table 3.** Results of the analysis of potential drivers of $\delta_v$ for all data points ('all times') and for the period with green unfolded leaves ('green leaves'). For each correlation the sign and the $R^2$ value is given in combination with the significance levels, marked with (*) for $p < 10^{-5}$, (*) for $p < 10^{-10}$, (*) for $p < 10^{-15}$ and so on.

| | $\delta D_v$ | | $\delta^{18}O_v$ | |
| | all times | green leaves | all times | green leaves |
|---|---|---|---|---|
| IF $\delta D$ | ⊕ 0.03 | ⊖ 0.12 * | ⊕ 0.03 | ⊖ 0.12 |
| IF $\delta^{18}O$ | ⊕ 0.01 | ⊖ 0.17 * | ⊕ 0.01 | ⊖ 0.17 * |
| $\frac{d\delta^{18}O_v}{dt}$ iso | ⊖ 0.06 | ⊖ 0.23 * | ⊖ 0.05 | ⊖ 0.20 * |
| $\frac{dD_v}{dt}$ iso | ⊖ 0.03 | ⊖ 0.17 | ⊖ 0.02 | ⊖ 0.16 |
| Temperature (44 m). | ⊕ 0.50 ******* | ⊕ 0.25 * | ⊕ 0.50 ******* | ⊕ 0.26 ** |
| Temperature (2 m) | ⊕ 0.52 ******* | ⊕ 0.27 ** | ⊕ 0.52 ******* | ⊕ 0.28 ** |
| Soil temperature (-2 cm) | ⊕ 0.45 ****** | ⊕ 0.18 * | ⊕ 0.43 ****** | ⊕ 0.20 * |
| Soil temperature (-64 cm) | ⊕ 0.23 ** | ⊕ 0.07 | ⊕ 0.19 ** | ⊕ 0.06 |
| LWUR | ⊕ 0.50 ******* | ⊕ 0.25 * | ⊕ 0.50 ******* | ⊕ 0.27 ** |
| LWDR | ⊕ 0.15 * | ⊕ 0.07 | ⊕ 0.16 * | ⊕ 0.12 |
| PARD | ⊕ 0.23 ** | ⊕ 0.01 | ⊕ 0.23 ** | ⊕ 0.01 |
| PARU | ⊕ 0.11 * | ⊕ 0.00 | ⊕ 0.11 * | ⊖ 0.00 |
| RH | ⊖ 0.27 *** | ⊖ 0.10 | ⊖ 0.26 *** | ⊖ 0.06 |
| VPD | ⊕ 0.30 *** | ⊕ 0.16 * | ⊕ 0.29 *** | ⊕ 0.13 |
| log ($C_{H2O}$) | ⊕ 0.35 **** | ⊕ 0.12 | ⊕ 0.36 **** | ⊕ 0.17 * |
| $C_{H2O}$ | ⊕ 0.37 ***** | ⊕ 0.16 * | ⊕ 0.38 ***** | ⊕ 0.21 * |
| $\delta D_{ET}$ | ⊕ 0.39 ***** | ⊕ 0.17 * | ⊕ 0.37 **** | ⊕ 0.16 * |
| $\delta^{18}O_{ET}$ | ⊕ 0.11 * | ⊕ 0.09 | ⊕ 0.11 * | ⊕ 0.09 |
| TKE | ⊖ 0.00 | ⊖ 0.03 | ⊖ 0.00 | ⊖ 0.03 |
| $u^*$ | ⊖ 0.01 | ⊖ 0.04 | ⊖ 0.00 | ⊖ 0.04 |
| $h$ | ⊕ 0.13 * | ⊕ 0.00 | ⊕ 0.13 * | ⊖ 0.00 |

dependencies indicate, that the temperature close to the surface is an important driver of $\delta_v$. In general, the correlation between temperature and $\delta_v$ might be linked to temperature dependent fractionation at the sites of evaporation. However, the day-to-day-variability is not fully reflected by the obtained correlations of $\delta_v$ with temperature and temperature-related quantities such as





LWDR, VPD and RH (see Table 3). This becomes clearer, when correlations between $\delta_v$ and its potential drivers are calculated

only for the time period between leaf unfolding and leaf senescence in fall (period: 'green leaves' in Table 3). For this time period, the obtained correlations with temperature-related quantities get weaker. The correlation with temperature at 2 m height for example is still significant ($p < 10^{-10}$) but has $R^2$-values of only 0.27 and 0.28 for $\delta D_v$ and $\delta^{18}O_v$, respectively. In general, the positive correlation with temperature related quantities implies that fractionation and evaporation might be relevant drivers of $\delta_v$. As we did not find indications that local ET drives the seasonal variability of $\delta_v$, the dominant source of the measured

water vapor might be further away, but the temperature during evaporation might still be correlated to the temperature during the measurement.

## 4  Conclusions

Here we evaluate laser spectroscopic measurements of the isotopic composition of water vapor ($\delta D_v$ and $\delta^{18}O_v$) in the SBL and local ET ($\delta D_{ET}$ and $\delta^{18}O_{ET}$) in combination with meteorologic and turbulence-related quantities on diurnal to seasonal

time scale above a managed beech forest in central Germany. We find that the temporal variability of the isotopic composition of water vapor in the SBL, even if measured closely above a managed beech forest, is not dominated by local ET at both, diurnal and seasonal time scale. Based on directly measured $\delta_{ET}$ and PBL height $h$ extracted from reanalysis data, we present an estimate of isoforcing-related change in the isotopic composition of ambient water vapor, $d\delta_v/dt_{iso}$. A direct comparison of measured $d\delta_v/dt$ with isoforcing-related $d\delta_v/dt$ reveals that local ET does not dominate the diurnal cycle of $\delta_v$ throughout

the year. We conclude that the diurnal cycle of $\delta_v$ is alternately dominated by local ET and entrainment, in particular in spring and summer. At seasonal time scales, we find no indication that local ET and entrainment dominate the observed variability. This could be related to a mutual canceling out of entrainment and local ET at this field site, yielding a more complicated seasonal variability. Further, the variability of $\delta_v$ over the full growing season can only partly be explained by Rayleigh distillation (linked to approximately 35 % of the variability). This fraction further decreases for the period when

green leaves were present and transpiration is expected. A larger fraction of 50 % of the observed seasonal variability of $\delta_v$ is linked to temperature, indicating some influence of temperature-related processes, such as fractionation during evaporation from different water pools. We conclude that the simultaneous measurement of $\delta_v$ and $\delta_{ET}$ in combination with meteorological and turbulence-related quantities and PBL height $h$ seems a promising approach to increase the understanding of the temporal variability in $\delta_v$. As the influence of local ET on $\delta_v$ is strongly moderated by boundary layer processes, accounting for PBL

height is particularly essential to understand changes in $\delta_v$.

## 5  Data availability

All data used for the figures presented here is provided in the supplementary material.





# 6 Author contributions

The isotope related research presented here was planned, carried out, described and interpreted by Jelka Braden-Behrens,
supervised by Alexander Knohl. The reanalysis dataset of the planetary boundary layer was extracted, analyzed and described
by Lukas Siebicke. All authors read and made editorial comments to the manuscript.

# 7 Competing interests

We declare that we have no conflict of interest.

# 8 Acknowledgements

This project was partly funded by the Dorothea-Schlözer-Fellowship and by the German Research Foundation (DFG, project
ISOFLUXES KN 582/7-1). Additionally this work was supported by the European Research Council via the European Union's
Horizon 2020 research and innovation programme (grant agreement no. 682512-OXYFLUX). We thank Dave Bowling for
reading and commenting on the manuscript.





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





## Appendix A:  Uncertainties of the PBL height

400   For 90% of the PBL height $h$, the absolute ensemble uncertainty was less than 122 m, corresponding to a relative ensemble uncertainty of less than 31% of the PBL height $h$, with lower relative uncertainties for higher PBL heights (see Fig A1) in the appendix. The mean uncertainty of the PBL height $h$ during 2016 was 6.6% of PBL height $h$, obtained as the slope of a major axis linear regression of the uncertainty of $h$ versus $h$ itself. Thus, the average uncertainty of 6.6% is small compared to the typically large diurnal changes of the boundary layer height. Given above uncertainty estimates of the reanalysis data, it is

405   appropriate to estimate boundary layer height based on reanalysis data.

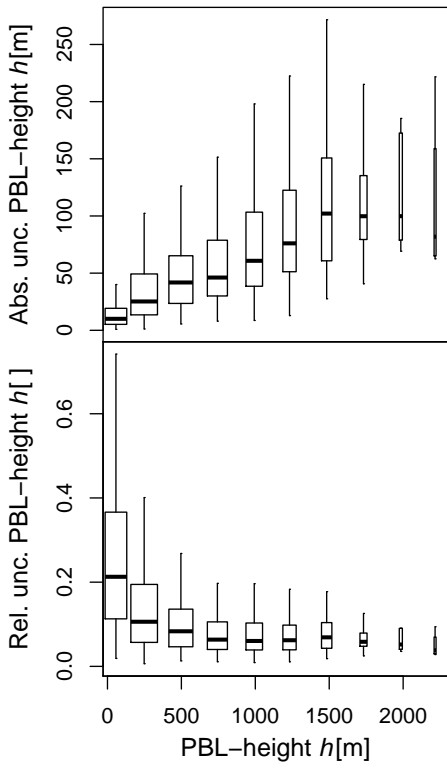

**Figure A1.** Box-whiskers plots for the uncertainties of the PBL height $h$. Top panel: Absolute uncertainty and bottom panel: relative uncertainty. The absolute uncertainty represents the ensemble spread in the reanalysis product and accounts for random errors. The relative uncertainty is absolute uncertainty normalized by the boundary layer height.