# Peer review of "Drivers of the variability of the isotopic composition of water vapor in the surface boundary layer"

_Biogeosciences, 2020_

## Referee Comment (RC1) · Anonymous Referee #1 · 8 Dec 2020

General comments: This manuscript presents the isotope ratio data of atmospheric water vapor (d18Ov and dDv) above a managed beech forest in central Germany. Together with EC measurements, values of d18O and dD associated with ET fluxes (dET) were also reported for a full growing season. The primary objectives of the study are to assess factors that are responsible for the observed variation in d18Ov and dDv. The authors used a simple linear regression to seek for correlations between d18Ov and dDv variability and an isolated variable and interpret their results on the basis of regression statistics (R2 and p-values). As far as I can tell, the experiment was properly carried out and the data were carefully scrutinized and high quality. The topic is interesting to a broad audience especially to the stable isotope community and researchers

who study ecohydrology. The measurements will contribute to a growing number of water vapor data collection, though a mechanistic interpretation of water vapor isotope data in the surface boundary layer remains challenging. The biggest issues I have with this manuscript are on the structure, assumptions made for the proposed problem, its statistical analysis and the interpretation of linear regression. The authors should consider addressing these major comments before its final publication.

Specific comments on major issues: 1. This manuscript has two major purposes, 1) to demonstrate the ET fluxes do not dominate d18Ov and dDv and 2) to evaluate potential factors that control d18Ov and dDv variability on both diurnal and seasonal time scales. For the 1st objective, the authors treat the PBL as a box, and assume surface ET is the only flux component that contributes to volume of the box (or the diurnal evolution of boundary layer height) while neglecting horizontal advection and entrainment fluxes. They used 'isoforcing' associated with the ET fluxes, combined with PBL heights retrieved from ECMWF data product, to calculate dv/dt (for both 18O and D) over the course of a day (Eq 2) and compared the results to the diurnal pattern from the time series measurements (Fig 2). Applying Eq 2 in this context is flawed as the height of the PBL cannot grow without entrainment (even if horizontal advection can be assume negligible under certain conditions) . Assuming surface flux as the only flux component while applying a changing height (h) contradicts one another. This is a misinterpretation of the boundary layer budget, and it is no surprise that calculated and measured dv/dt diurnal pattern has nothing in common (Fig 2). Secondly, why would the authors even bother to do this exercise? As later stated by the authors (ln 189-190) "A discussion of the influence of local ET that is purely based on isoforcing IF overlooks the influence of boundary layer mixing processes." The authors later stated that "the concurrent trends in the diurnal cycles of CH2O and dv indicate, that entrainment dominantly influences dv from the forenoon to the afternoon:" and ". . . we observe this indication for a dominant influence of entrainment from the forenoon to the afternoon also in summer." If you can make these conclusions from the observation (which the authors did), why trying to prove (and did it incorrectly) something that the data have

already shown? This whole section should be scratched in my view. 2. For the second objective, the authors identify 4 potential factors that influence seasonal availability of dv: local ET, Rayleigh distillation, selective water use by plants and temperature. The author applied a simple linear regression between dv and each of these factors to look for correlations. There are several problems in the statistical analysis used by the authors. First, the authors should distinguish processes from state variables. Secondly, these factors are not independent from one another, for example, ET and Rayleigh distillation are both temperature dependent. A simple linear regression ignores the interactive effect between processes and state variables. Ideally, one should carry out a full BL budget calculation with a numeric model that considers thermodynamic isotopic fractionation. At the very least, the authors need to consider a multivariate regression that considers the interactive effect among variables. A simple linear regression is inappropriate. 3. The authors use sloppy statistics. This manuscript reports incredibly small p-values ($10^{-35}$) that are simply not meaningful. The p-value is calculated from the data and depends on the sample size (number of data points). It is possible to get p values to the -35 decimal points but that is simply because of we have the computing power to do this. More data points give you smaller p values. The bigger issue is, is the p value reliable? The p values shown in Table 3 are simply not meaningful. The difference in the p values between all times and period of green leaves is likely an artifact of sample size. Some statisticians have urged not to use p values but to use other alternative statistical matric because it is too often misinterpreted (see Halsey 2019 http://dx.doi.org/10.1098/rsbl.2019.0174). This study is another example of why. The authors should limit reporting p values to a more reliable estimate. 4. After redo the statistical analysis, the authors must re-evaluate their interpretation of the results and draw proper conclusion accordingly.

Technical comments: Ln 25. Do you mean a major driver of dv variability? Why the remove of precipitation only acts on seasonal time scales? Ln29. Your description of the amount effect is very crude and can cause confusion. Please be more elaborative on the amount effect. Ln37-38. It's unclear what 'different importance' means based on

R2 values; this sentence is hard to read. It's easier to see the effect by a state variable (such as temperature) but it becomes harder to visualize by a process (like Rayleigh rainout). Can you explain how Rayleigh process may differ seasonally that in turn affect seasonal variability of dv? Ln51-52, a correlation does not suggest a causal effect; maybe that was not what you meant to suggest but the writing makes it seem that way. Ln61-62, consider revise this sentence to "Only one of these studies performed direct dET measurements in a forest". Ln99 pls provide more details on how exactly dET was calculated. Did you perform a spectral analysis to examine potential loss of energy due to the differences in the sampling frequency between EC and isotope measurements? Ln123 More precisely speaking, VPD is calculated from temperature and RH data which were directly measured. Ln128. Can you give a brief description on how the rain sampler is designed to store its water to prevent evaporation? Ln135. avoid jargon; just say using ECMWF data product Ln145 delete this sentence; rework this paragraph. Rather than copying from the manual, it would be more useful to describe how you retrieve PBL h from IFS. Ln179 what is the time unit here? Is this 0.1 permil per second, per minute or per hour? Assuming 0.1 permil per hour, from 8am to 5pm, dv would've increased by 0.9 permil d18O and almost 10 permil dD. But Fig 1 shows a decrease in d18O by $\sim$ 1 permil while a decrease in dD by $\sim$ 5 permil. How do you reconcile the inconsistency between these results? Ln180-185 As stated above, this conclusion is flawed as the calculation was based on an invalid assumption of no entrainment while BL h is allowed to grow. Ln186-194 These remarks acknowledge the authors have known the answer from the observation but still decided to use a reverse logic to disapprove something they already knew could not be true. Hmmm interesting . . . Ln195-200 I found this section puzzling. TKE is a measure for the intensity of turbulence. h is most commonly defined by an inversion in potential temperature and dewpoint and is often estimated by radio sounding or lidar. It does not make sense to make direct comparison between TKE to PBL h (yes, they are both part of the boundary layer dynamics) because there is not a causal effect between the two. Simply presenting correlations without context is meaningless (if seeking covariation is the goal, why

not presenting correlations with other meteorological variables? why do you choose to only present TKE? I would suggest removing TKE altogether. Ln204 -12 permil for dDv? Is this a typo? Ln206 Shouldn't selective water use by plants be included in ET? Ln209 some would argue 7m above the top of the canopy is pretty far out; it is likely outside the subsurface BL near the forest canopy. Since you mention TKE, why don't you show a profile of vertical wind speed and momentum fluxes? It will give you an idea if your sensors are within the canopy subsurface BL. Ln222-223, do you have an explanation of why you found a correlation between dET and dDv but not wit d18Ov? Ln225-234 these interpretations are based on flawed stats Ln238. Bowling et al. 2017 is not an appropriate citation here. Remove. Fig 7. Right panel: after leaf coloring - was that diamond or cross symbol? Ln243-248 & Fig7. Are GMWL and LMWL statistically different? I am skeptical that you can use GMWL and LMWL to contrast impacts by far-field v.s near-field factors. Ln254. entrainment is a diurnal process. Why would you expect entrainment be a factor on seasonal time scales in the first place?

---

## Referee Comment (RC2) · Anonymous Referee #2 · 10 Dec 2020

In this manuscript, the authors present a dataset of the isotopic composition of water vapor over a forested ecosystem. They combine the measurements of the water vapor with eddy covariance derived estimates of the isotopic ratio of the ET flux. The goal was to test a fundamental hypothesis that the isotopic ratio of water vapor above an actively transpiring surface should respond to the ET flux. Over large scales (i.e. from satellite data) it has certainly been shown that the land surface fluxes of water vapor influence the isotopic ratio of the atmospheric water vapor. The authors conclude that the ET flux has minimal influence on the isotopic ratio of vapor. On diurnal timescales, entrainment drives a midday depletion in the water vapor isotopes that is in opposition to the flux of ET. On seasonal timescales, entrainment rate does not predict the isotopic ratio of the

vapor. Rather, it is some combination of processes (ET, rainout and temperature) that collectively influence the isotopic ratio of the vapor.

Firstly, I commend the authors on a very nicely developed dataset and some rather sophisticated analysis of the data. Secondly, I think the question that is posed is interesting and worthwhile particularly to the extent that using water isotopes to trace water fluxes and close hydrological budgets in the atmosphere has a lot of potential in diagnostic analysis of GCMs and transport models. This work contributes to these efforts. However, I found the analysis, on the one hand, to be unnecessarily complex at times (i.e. there were many competing correlations between derivatives) but also overly simple at others (i.e. trying to use a single linear regression model to predict d18Ov). In the end, I think the authors overlooked some simple tests that could have been useful and drew conclusions regarding why d18O/dDv correlated with temperature that are not correct. I would support publication after significant changes are made to the writing and perhaps some additional analyses.

The authors find that entrainment is the prominent driver of the diurnal cycle in d18Ov except in the morning when transpiration has more of an effect. This finding has been very clearly identified in previous works. See for example: doi.org/10.1002/jgrd.50701 as well as numerous other citations the authors provide. It would seem therefore that the authors should not have been surprised to find this to be true. It would have been surprising, in fact, to find the opposite to be true. I this comment is significant because it affects the entire tone of the paper. The authors should have begun from the perspective that entrainment is the primary driver of diurnal cycles and then sought examples where the effect of ET emerged.

The authors find that the correlation between entrainment rate and the seasonal cycle in d18Ov is weak. They therefore conclude that entrainment is not the critical driver of the seasonal cycle. However, they fail to identify that it is not just how much vapor is entrained but the isotopic ratio of the water vapor that is entrained. With synoptic scale changes in atmospheric circulation the isotopic ratio of water within the free troposphere changes. It would seem quite clear, and maybe I misunderstood this from the manuscript, that it is the isotopic ratio of the free troposphere driven by large scale circulation that drives changes in the midday isotopic ratio above the canopy. Analysis using a lagrangian transport could be deployed (as with many previous isotope studies) to identify how the source of vapor changes and whether it is the source region that explains the seasonal changes.

The authors find a strong influence of temperature on d18Ov and call upon a rather confusing role for temperature influencing the fractionation of ET. I find this extremely unlikely. If this was the case, then there should be a very strong relationship between deltaET and temperature. I believe deltaET is more strongly influence by RH or VPD and or LAI. Revisiting comment #3, changes in synoptic circulation drive both changes in temperature and the d18Ov. The temperature of air masses affect how much rainout has occurred and give rise to a strong relationship between d18Ov and temperature. This is in fact the rationale for why ice core d18O values reflect temperature. I think explaining the relationship between d18Ov and temperature would have benefited from taking a more "first principles" approach and yielding to extensive research already done on this topic.

The calculation of isoforcing relied heavily on the estimates of PBL height from reanalysis. This concerned me somewhat because there was no good validation of these estimates and it seems the estimates from reanalysis would only be useful if the land cover in the area was homogenous. In other words, is the forested cover of the site representative of the conditions with the reanalysis grid cell? The authors discuss error estimates of PBL height but it was not clear how these error values were assimilated in the analysis. Secondly, the authors note that their assumption that the isotopic ratio of water vapor is well mixed is likely incorrect. This has been shown by other studies using gradient and flux gradient approaches. What are the effects of this assumption on the isoforcing estimate? What if the authors assumed a gradient with log form up the top of the PBL using previous studies? My point is that if the authors know this

assumption is incorrect it would be valuable to assess the impact of this assumption on their analysis using a sensitivity approach.

I was surprised come to the end of the paper and never see a figure or actual discussion on the estimates of delta ET. The estimates of delta ET were assimilated into numerous analyses but, after all, if the study is looking at how delta ET affects delta V, the readers should see delta ET. The authors need to present this data and analyse it directly before using it in more sophisticated approaches. How does delta ET vary through the season? Was it affected by soil moisture and VPD that might change T and E partitioning? Did delta ET relate to total ET rates or greeness/LAI? Does it change after rainfall events? An analysis of the drivers of delta ET are a necessary complement to the other analyses presented.

Small comments:

When a variable is introduce the correct grammar (I think) is like this. . . "Temperature, T, is related to latitude." Or "Temperature (T) is related to latitude."

28: Unclear why sublimation of snow was listed under "precipitation removal" processes. This would be a surface flux process.

54: The R2 value between C and d18O/dD were just listed in the previous paragraph so this sentence felt redundant.

60: Lots of other studies over forests not considered here: Continuous measurements of atmospheric water vapour isotopes inwestern Siberia (Kourovka) Stable Water Isotopes Reveal Effects of Intermediate Disturbance and Canopy Structure on Forest Water Cycling Response of water vapour D-excess to land–atmosphere interactions in a semi-arid environment I would say broadly that the literature available on this topic was under-cited.

145-155: This extended quotation from ERA5 manual is not appropriate. The authors should explain the process of error estimation in their own words. As noted above, it is

also unclear how this error was assimilated in the analyses that follow.

163: Missing "space" before the sentence begins.

170 "site" not "cite"

171: "However" is the wrong word here because this sentence does not contradict the previous one it supports it.

176: The comma should be after "h" not after "both"

178: if the nighttime data is not meaningful, I would recommend excluding it. As you note, when the value approaches 0, the equation becomes very unstable.

181: When you write ddv/dt is this dt_iso or dt_meas. Truthfully, I found the comparisons between the many derivatives quite hard to follow and perhaps not the most useful way to analyze the dataset. Figure 1: Standard error should be reported around composite diurnal cycles.

193: "being"

206: I was confused as to what the authors mean by Rayleigh distillation in this context. Is this condensation onto the surface such as through dew or is this the collective rainout of the air mass as it ages from its origin?

206: Also, because all of these processes are important to the hydrological balance, it would seem that linear univariate models are not really appropriate or useful. Perhaps multivariate non-linear models would be better suited for partitioning the relative controls.

207: "between"

208: missing closed parenthesis at end of paragraph.

209: Earlier you discuss the inlet being 10 m above canopy but here you say 7 m. Not a big deal but better to just be consistent.

Figure 6 and associated discussion on Rayleigh Distillation: The assumption that a single distillation model (i.e. a linear fit to d18O vs. log(C)) assumes that a common source but experiencing different degrees of rainout. This is not true. So you could really have multiple plausible distillation models that would give rise to "messier" scatter plot of your data.

253: How does delta ET relate to precipitation? This could give you some insight into the fractionation of ET relative to the source. Does it change during the year?

261-262: The authors write: "In general, the correlation between temperature and v might be linked to temperature dependent fractionation at the sites of evaporation." What are the sites of evaporation being referred to here? Local ET? Nearby lakes that might supply atmosphere? The ocean source?

---

## Referee Comment (RC3) · Anonymous Referee #3 · 21 Dec 2020

This manuscript deserves final publication in BG after a few minor corrections and editorial adjustments. The authors present their analysis in a clear and logical fashion. The dataset was obtained with a well-tested instrumental system. A key strength of this analysis is the measurement of the vapor isotopic flux to inform interpretation of physical drivers of the observed vapor isotopic variability.

Line 150: Unlike other independent variables, here the PBH height is model-derived. Can you comment on efforts (by you or others) to evaluate the ERA h against observed h for your geographic region?

Line 190: The message here is quite clear. Can you comment on the implication for

[Figure]

Keeling mixing line analysis?

Line 207: typo "betreen"

Line 215: you mean "... when we expect NO transpiration..."?

Line 233: Some people consider the lack of correlation between vapor delta and concentration as indicative of Rayleigh distillation associated with atmospheric convection. (When an air parcel movement span a large vertical distance, condensation occurs over a large range of temperature.)

Figure 1: ET unit is incorrect. The unit carried by IF is different from that shown in Figure 4

Figure 7 left panel: I don't see rain data

Figure 7 right panels: These basically reveal seasonal pattern of vapor d-excess. Can you comment on diurnal pattern of vapor d-excess and its implications?

Figure 2 & Table 3: How did you obtain TKE?

---

## Author Comment (AC1) · 19 Feb 2021

**AUTHORS COMMENT: ANSWER TO REFEREE 1**
**'Drivers of the variability of the isotopic composition of water vapor in the surface boundary layer'**

**Referree comments:** black,
**Author comments:** blue
**Changes to the manuscript:** green

General comments: This manuscript presents the isotope ratio data of atmospheric water vapor (d18Ov and dDv) above a managed beech forest in central Germany. Together with EC measurements, values of d18O and dD associated with ET fluxes (dET) were also reported for a full growing season. The primary objectives of the study are to assess factors that are responsible for the observed variation in d18Ov and dDv. The authors used a simple linear regression to seek for correlations between d18Ov and dDv variability and an isolated variable and interpret their results on the basis of regression statistics (R2 and p-values). As far as I can tell, the experiment was properly carried out and the data were carefully scrutinized and high quality. The topic is interesting to a broad audience especially to the stable isotope community and researchers who study ecohydrology. The measurements will contribute to a growing number of water vapor data collection, though a mechanistic interpretation of water vapor isotope data in the surface boundary layer remains challenging. The biggest issues I have with this manuscript are on the structure, assumptions made for the proposed problem,its statistical analysis and the interpretation of linear regression. The authors should consider addressing these major comments before its final publication.

**Authors response:** We thank the anonymous referee for the motivating, detailed and constructive feedback to our manuscript, below we answer the referee's comments in detail.

Specific comments on major issues:
1. This manuscript has two major purposes, 1) to demonstrate the ET fluxes do not dominate d18Ov and dDv and 2) to evaluate potential factors that control d18Ov and dDv variability on both diurnal and seasonal time scales. For the 1st objective, the authors treat the PBL as a box, and assume surface ET is the only flux component that contributes to volume of the box (or the diurnal evolution of boundary layer height) while neglecting horizontal advection and entrainment fluxes. They used 'isoforcing' associated with the ET fluxes, combined with PBL heights retrieved from ECMWF data product, to calculate dv/dt (for both 18O and D) over the course of a day (Eq 2) and compared the results to the diurnal pattern from the time series measurements (Fig 2). Applying Eq 2 in this context is flawed as the height of the PBL cannot grow without entrainment (even if horizontal advection can be assume negligible under certain conditions). Assuming surface flux as the only flux component while applying a changing height (h) contradicts one another. This is a misinterpretation of the boundary layer budget, and it is no surprise that calculated and measured dv/dt diurnal pattern has nothing in common (Fig 2).

**Authors response:**
We thank the anonymous referee for pointing this out. We acknowledge, that the current version of the manuscript might not be sufficiently clear about the purpose, the assumptions, and the limits of this estimation.

We are well aware, that the calculation of $\frac{d\delta_v}{dt}|_{ET,est}$ based on equation (2) does not yield a real estimate of $\frac{d\delta_v}{dt}$ and in particular assuming a temporarily constant PBL height is only a theoretical assumption that is not justified for longer timescales. We want to clarify that temporarily changing PBL height changes the isotopic composition in two ways: 1) Entrainment of isotopically different

material from higher layers and 2) Changes in the relative fraction between the gas masses (i.e. dilution of the isoforcing signal over a larger volume). For our thought experiment we assume entrainment to not directly change $\delta_v$ by different material but only allow an influence of $\delta_v$ by dilution. The influence of local ET on $\delta_v$ is diluted by different PBL heights at different times of the day, thus throughout the day, IF/h reflects the influence of ET on $\delta_v$ in a boundary layer with a certain (slowly changing) height.

With the calculation based on equation 2, we quantify the influence of local ET on the isotopic composition of the boundary layer by making a quantitative thought experiment. More specifically, we do not aim at fully modelling $\delta_v$ (which is beyond the scope of our manuscript), but we want to answer the following question: How would local ET influence the delta value of the PBL ($\delta_v$ ) if local ET would be the only process that (significantly) influences $\delta_v$ . More particular:  We want to quantify/identify the influence of local ET for the theoretical case, that throughout the day entrainment would change PBL height as observed, but would not change $\delta_v$ . By doing so, we isolate the influence of local ET on $\delta_v$ . In particular, this reflects the influence of local ET better than assuming a constant PBL height and we find evidence that just using IF values alone is inappropriate to conclude about the influence of local ET on $\delta_v$ . This is discussed in Line 224 FF of the revised manuscript when we discuss possible interpretations of the diurnal cycle of isoforcing and in Line 267 FF, when we discuss that correlations between $\delta_{v \text{ and }} \delta_{ET}$ should not be over-interpreted.

Thus, this approach, despite its limitations, yields a quantitative estimate for isoforcing-related changes in $\delta_v$ which is more appropriate than directly using isoforcing values or using isoforcing values in combination with assuming a PBL height that is constant throughout the day. For this reason, we would like to keep this approach in the manuscript, but we revised the manuscript to be clearer about the purpose and the limitations of this approach, in particular, we rewrote section 2.4 of the manuscript. We changed the manuscript to explicitly mention that the quantity, which we now call $\frac{d\delta_v}{dt}|_{ET,est}$, is not the real change in delta value, but only a theoretical estimate of the influence of ET, which is only one out of many. In the revised manuscript we changed section 2.4 about the Calculation of evapotranspiration-related change in $\delta_v$ , to be clearer about about the purpose, the assumptions and the limits of this estimation. Further, we differentiate strictly between d $\delta_v$ /dt_meas and $\frac{d\delta_v}{dt}|_{ET,est}$ to make clear that these two quantities are not the same.

**2.4 Calculation of evapotranspiration-related change in $\delta_v$**

We quantify the influence of local ET on the isotopic composition of the boundary layer by making a quantitative thought experiment. How would local ET influence the delta value of the PBL ($\delta_v$ ) if local ET would be the only process that (significantly) influences $\delta_v$? To answer this question, we use isoforcing values, that are based on EC measurements of the magnitude of ET $F_{ET}$ and its  isotopic composition $\delta_{ET}$ (see Braden-Behrens2019). We further assume a simple isotopic mass balance model (see e.g. Lai2006) with only one flux component (ET) from the surface and no influence of horizontal advection or entrainment on $\delta_v$ (see alsoSturm2012,Braden-Behrens2019). If this assumption would be fulfilled, isoforcing IF can be interpreted as the rate of change of the atmospheric delta value multiplied by the temporarily constant boundary layer height h (see e.g. Lai2006).

$$\text{IF} = \frac{F_{ET}}{C_a \rho_a}(\delta_{ET} - \delta_v) = h\frac{d\delta_v}{dt}|_{ET,est}$$

$$\Rightarrow \frac{d\delta_v}{dt}|_{ET,est} = \frac{IF}{h}$$

(Eq. 2 and 3)

With the evaporative flux F_ET, its isotopic composition $\delta_{ET}$ the atmospheric mole fraction $C_a$, the molar density of atmospheric air rho$_a$, the atmosphere's isotopic composition $\delta_v$ and the height h of the planetary boundary layer (PBL).

We use Eq. \refeq:isoforcing2 to calculate $\frac{d\delta_v}{dt}|_{ET,est}$ for our measurements at different times of the day with a simultaneous estimation of the PBL height for each data point. As evident from Eq. 3 the influence of local ET on $\delta_v$ is diluted by different PBL heights h. Thus in particular as h changes throughout the day, $\frac{d\delta_v}{dt}|_{ET,est} = IF/h$ reflects the influence of ET on $\delta_v$ in a boundary layer with a certain (slowly changing) height. The resulting quantity $\frac{d\delta_v}{dt}|_{ET,est}$ yields a theoretical estimate for the influence of local ET on $\delta_v$. However, the real change of $\delta_v$ is composed of changes related to many different drivers such as entrainment or horizontal advection see e.g.Griffis2007.

Secondly, why would the authors even bother to do this exercise? As later stated by the authors (ln 189-190) "A discussion of the influence of local ET that is purely based on isoforcing IF overlooks the influence of boundary layer mixing processes."

**Authors response:** Our sentence was not clear enough in the original manuscript. We referred to a discussion of the impact of ET based on isoforcing values IF versus a discussion based on $\frac{d\delta_v}{dt}|_{ET,est}$ (which includes a changing PBL-height). We do not refer to a general discussion of drivers of $\delta_v$, but more specifically on the role of h when calculating the impact of local ET on measured $\delta_v$. This has sometimes been discussed and estimated with assuming a PBL height that is constant on longer timescales. We change this sentence and add a discussion of the diurnal cycle of isoforcing: 'Our data shows that a discussion of the influence of local ET that is purely based on isoforcing IF and does not include PBL height yields an over/underestimation of $\frac{d\delta_v}{dt}|_{ET,est}$. If we simply would assume a constant PBL height of eg. 1km, we would underestimate the influence of local ET for most of the times except around midday in spring and autumn. Further, if we would have used the diurnal cycles of isoforcing (see Fig. 1) as an indication for the influence of ET on $\delta_v$ throughout the day, we would have concluded that ET has the strongest influence on ET around midday. Our estimation of $\frac{d\delta_v}{dt}|_{ET,est}$ on the other hand shows a comparable magnitude in the mornings and in the evenings, while the comparison to $\frac{d\delta_v}{dt}$ shows that $\delta_v$ is dominantly driven by other processes such as entrainment around midday. Thus, we further conclude that due to the large variability of the boundary layer height h, it is essential to account for h when estimating the influence of local ET on ambient water vapor.'

The authors later (LINES 190 FF) stated that "the concurrent trends in the diurnal cycles of CH2O and dv indicate, that entrainment dominantly influences dv from the forenoon to the afternoon:" and ": : : we observe this indication for a dominant influence of entrainment from the forenoon to the afternoon
also in summer." If you can make these conclusions from the observation (which the authors did), why trying to prove (and did it incorrectly) something that the data have already shown? This whole section should be scratched in my view.

**Authors response:** We still would like to keep the quantification of $\frac{d\delta_v}{dt}|_{ET,est}$, because we think it is more convincing to show both: a) a direct but only qualitative **indication** for the influence of entrainment by simply interpreting the shape of the diurnal cycles of C and $\delta_v$ and b) the quantitative estimation of $\frac{d\delta_v}{dt}|_{ET,est}$ in comparison with the measured $\frac{d\delta_v}{dt}$. This way, we can identify the magnitude of isoforcing related change in $\delta_v$.

2. For the second objective, the authors identify 4 potential factors that influence seasonal availability of dv: local ET, Rayleigh distillation, selective water use by plants and temperature. The author applied a simple linear regression between dv and each of these factors to look for correlations. There are several problems in the statistical analysis used by the authors. First, the authors should distinguish processes from state variables. Secondly, these factors are not independent from one another, for example, ET and Rayleigh distillation are both temperature dependent. A simple linear regression ignores the interactive

effect between processes and state variables. Ideally, one should carry out a full BL budget calculation with a numeric model that considers thermodynamic isotopic fractionation. At the very least, the authors need to consider a multivariate regression that considers the interactive effect among variables. A simple linear regression is inappropriate.

**Authors response:** Thanks for this remark. We agree that the statistical analysis benefits from a multivariate regression. A full BL budget calculation that includes thermodynamic isotopic fractionation would be beyond the scope of this work. In the revised manuscript, we present a multivariate regression of the dataset.

3. The authors use sloppy statistics. This manuscript reports incredibly small p-values ($10^{-35}$) that are simply not meaningful. The p-value is calculated from the data and depends on the sample size (number of data points). It is possible to get p values to the -35 decimal points but that is simply because of we have the computing power to do this. More data points give you smaller p values. The bigger issue is, is the p value reliable? The p values shown in Table 3 are simply not meaningful. The difference in the p values between all times and period of green leaves is likely an artifact of sample size. Some statisticians have urged not to use p values but to use other alternative statistical matric because it is too often misinterpreted (see Halsey 2019 http://dx.doi.org/10.1098/rsbl.2019.0174). This study is another example of why. The authors should limit reporting p values to a more reliable estimate.

**Authors response:** Thanks for this comment and for pointing out the interesting paper by Halsey et al. 2019. We fully agree that reporting such small p-values is not helpful. We will correct this and will use a p-value notion marking only ($p<10^{-5}$) with a * and also give AIC numbers for multivariate regressions.

4. After redo the statistical analysis, the authors must re-evaluate their interpretation of the results and draw proper conclusion accordingly.

**Authors response:** We added a multivariate regression and changes the discussion accordingly. Concerning ET as a potential driver, we find a negative dependency between $\frac{d\delta_v}{dt}|_{ET,est}$ and $\backslash \delta_v$ also in the multivariate regression. We discuss that this is physically not meaningful. Further, the multivariate regression did not yield a physically meaningful explanation with lower AIC than a simple correlation with temperature as the only driver. Thus, in the revised manuscript we focus more on this correlation.

Technical comments:
Ln 25. Do you mean a major driver of dv variability? Why the remove of precipitation only acts on seasonal time scales?

**Authors response:** Here we focus on Rayleigh distillation as a cumulative removal of rain from the atmosphere. We think this might not have been clear enough in the original manuscript. We changed this sentence to 'At seasonal time scales the cumulative rainout of an air mass as it ages from its origin (e.g by Rayleigh destillation) is a major driver of the variability of $\delta_v$.'

Ln29. Your description of the amount effect is very crude and can cause confusion. Please be more elaborative on the amount effect.

**Authors response:** We rephrased the writing to be clearer about the complexity of the empirical amount effect, that can be a result of many different processes depending on the location. We further added some information on the influence of deep convection on the amount effect, as mentioned by Tharammal et al. 2017 JGR-A: 'These complex processes yield the 'temperature effect', a positive correlation between condensation temperatures and higher delta-values of precipitation (see e.g. Dansgaard1964) and the empirical 'amount effect', a negative correlation between the total amount and the mean isotopic composition of precipitation (see e.g. Dansgaard1964, Tharammal2017). However, the 'amount effect' can be a result of many different processes

depending on the location. For example the amount effect can be strongly moderated by deep convection (see e.g. Tharammal2017).'

Ln37-38. It's unclear what 'different importance' means based on R2 values; this sentence is hard to read. It's easier to see the effect by a state variable (such as temperature) but it becomes harder to visualize by a process (like Rayleigh rainout). Can you explain how Rayleigh process may differ seasonally that in turn affect seasonal variability of dv?

**Authors response:**
With 'difference importance' we wanted to refer to the considerable differences in correlations between log(T) and $\delta_v$ that have been done by many other authors and that have been interpreted to reflect in how far the data could be explained by Rayleigh distillation. In the revised manuscript we explain this a bit more detailed: 'Thus, at different field sites, $\delta_v$ and log(T) are differently strong correlated. This indicates, that Rayleigh processes might play a dominant role in some cases (potentially explaining up to 78% of the variability) while in other cases other processes are more relevant (see also Huang2014 for details).'
We also add some more information and citations here to explain that Rayleigh distillation is only a very simple model for the cumulative removal of rain from the atmosphere: 'However, the removal of rain from the atmosphere by Rayleigh distillation is only a very simple model, while both, changes in the originating air masses and rainout processes are much more complex (see e.g. Noone2011).

Ln51-52, a correlation does not suggest a causal effect; maybe that was not what you meant to suggest but the writing makes it seem that way.

**Authors response:** We agree and changed the writing to not imply that a correlation suggests a causal effect. New version: 'At seasonal time scales, some authors found evidence for a dominant role of Rayleigh processes (Lee2006, Wen2010).'

Ln61-62, consider revise this sentence to "Only one of these studies performed direct dET measurements in a forest".

**Authors response:** We followed the suggestion and changed to 'Only one of these studies, the one by Huang2014, performed direct $\delta_{ET}$ measurements in a forest, however based on a flux-gradient approach, not eddy covariance.'

Ln99 pls provide more details on how exactly $\delta_{ET}$ was calculated. Did you perform a spectral analysis to examine potential loss of energy due to the differences in the sampling frequency between EC and isotope measurements?

**Authors response:** Thanks for this remark we agree that it is helpful to add some more details about the data evaluation to the manuscript – this might have been to short in the original manuscript. Concerning the measurement frequencies of the different instruments, we add: 'We combined the 20Hz anemometer measurements with the 2Hz measurements of C_H2O, $\delta^{18}O_v$ and $\delta D_v$ yielding a 2Hz dataset of simultaneous measurements of isotopologue concentrations and 3D windspeed to calculate the magnitude and the isotopic composition of ET using the eddy covariance software EddyPro, version 6.2.0 LiCorBiosciences2016.'
Concerning data evaluation steps for flux calculations, we added:
'The used method to correct for high-frequency dampening, was based on the work of Ibrom2007, as recommended for closed path analyzers with loge tubing (LiCorBiosciences2016).'
Concerning the influence of the reduces measurement frequency, we add the following to the revised manuscript: 'In particular, we analyzed the influence of technical limitation such as the comparably slow measurement frequency of 2 Hz on water vapor flux measurements by additionally using 20Hz measurements of $C_{H2O}$ using a standard closed path $CO_2$ and $H_2O\_v$ analyzer (LI-6262 LiCor Inc., Lincoln, USA). We mathematically reduced its measurement frequency down to 2Hz seeBraden-Behrens2019 and found that the resulting 2Hz dataset captured more than 98% of the variability of the 20Hz dataset (see Braden-Behrens2019).'
However, for a detailed description of the different and complex data evaluation steps, we refer to

our technical manuscript about EC measurements of $CO_2$ (Braden-Behrens2019).

Ln123 More precisely speaking, VPD is calculated from temperature and RH data which were directly measured.
**Authors response:** Thanks, for pointing this out, in the revised manuscript, we removed 'VPD' from this list, because it is not directly measured.

Ln128. Can you give a brief description on how the rain sampler is designed to store its water to prevent evaporation?
**Authors response:** Yes, we include the following to the manuscript:
'In brief, these rain samplers, reduce evaporation by minimizing the water surface exposed to the atmosphere. This is achieved by using a thin tube from the funnel down to the bottom of the sampling bottle and additionally using a very long and thin tube to adjust the air pressure in the sampling bottle, (see Groning2012).'

Ln135. Avoid jargon; just say using ECMWF data product
**Authors response:** We changed the whole section about PBL height and avoid jargon.

Ln145 delete this sentence DONE; rework this paragraph. Rather than copying from the manual, it would be more useful to describe how you retrieve PBL h from IFS.
**Authors response:** The whole section on PBL height has been reworked, replacing the quotes from the manual, and describing how PBL h has been retrieved from the IFS/ERA5 product. It should be clearer now that both PBL height as well as the associated random uncertainty is a product readily delivered as part of ERA5 rather than being derived in the current study. In addition, we have added information on the uncertainty of the product relative to radiosonde measurements and on the representativity of the grid cell of ERA 5 relative to the study site.

Ln179 what is the time unit here? Is this 0.1 permil per second, per minute or per hour? Assuming 0.1 permil per hour, from 8am to 5pm, dv would've increased by 0.9 permil d18O and almost 10 permil dD.  But Fig 1 shows a decrease in d18O by _ 1 permil while a decrease in dD by _ 5 permil. How do you reconcile the inconsistency between these results?
**Authors response:**
Yes, the unit is per hour. We added this missing unit to the manuscript. The inconsistency that you are referring to is the difference between $\frac{d\delta_v}{dt}|_{ET,est}$. This is exactly what we refer to in line 179ff – but instead of focusing on $\delta_v$ (Fig 1), we focus on its temporal derivative $\frac{d\delta_v}{dt}|_{meas}$ in Fig. 2. We address this discrepancy in the followingsentence: 'The directly measured diurnal cycles of $\frac{d\delta_v}{dt}|_{meas}$ do not agree with this isoforcing-related estimate $\frac{d\delta_v}{dt}|_{ET,est}$ (see Fig. 2). In particular in spring and summer, we measure negative values of $\frac{d\delta_v}{dt}|_{meas}$ around midday, associated with a depletion of ambient water vapor, while the isoforcing-related change $\delta_v$ always yields an enrichment.'
We think this is now clearer after we distinguished more consistently between $\frac{d\delta_v}{dt}|_{ET,est}$ and between $\frac{d\delta_v}{dt}|_{meas}$ and also changed the axis label in Fig 2 accordingly.

Ln180-185 As stated above, this conclusion is flawed as the calculation was based on an invalid assumption of no entrainment while BL h is allowed to grow. **Authors response:** Please see our comment above.

Ln186-194 These remarks acknowledge the authors have known the answer from the observation but still decided to use a reverse logic to disapprove something they already knew could not be true. Hmmm interesting

**Authors response:** There are different aspects that might have caused unclarity here:

1. Here we distinguish between isoforcing IF and $\frac{d\delta_v}{dt}$. (which is based on Isoforcing, but includes dilution by the PBL). Please see our comment above, referring to lines **189-190**.

2. As explained above (referring to line 190 ff), we draw our conclusions on the diurnal cycles shown in figures 1 and 2 because we think it is more convincing to show both: a) a direct but only qualitative **indication** for the influence of entrainment by simply interpreting the shape of the diurnal cycles of C and $\delta_v$ and b) the **quantitative estimation** of $\frac{d\delta_v}{dt}|_{ET,est}$ in comparison with the measured $\frac{d\delta_v}{dt}$. This way, we can identify the magnitude of isoforcing related change in $\delta_v$. We think using both approaches is much more convincing than only discussing the shape of measured C and delta values.

3. The focus of our analysis was on quantifying the influence of ET for different timescales. We try to be clearer about that in the revised manuscript. Eg. We added the following to the introduction: 'We hypothesize, that at our measurement position, local ET is an important driver of $\delta_v$ at both, the seasonal and the diurnal time scale and use our direct measurements in combination with PBL height h to quantify the influence of ET on $\delta_v$ .'

We additionally restructured the paragraph (lines 186FF of the original manuscript) because we think this help to be clearer about the reasons for our conclusions.

Ln195-200 I found this section puzzling. TKE is a measure for the intensity of turbulence. h is most commonly defined by an inversion in potential temperature and dewpoint and is often estimated by radio sounding or lidar. It does not make sense to make direct comparison between TKE to PBL h (yes, they are both part of the boundary layer dynamics) because there is not a causal effect between the two. Simply presenting correlations without context is meaningless (if seeking covariation is the goal, why not presenting correlations with other meteorological variables? why do you choose to only present TKE? I would suggest removing TKE altogether.
**Authors response:** We removed this section/ the analysis of TKE from our manuscript.

Ln204 -12 permil for dDv? Is this a typo?
**Authors response:** Yes, this was a typo. We changed it to to -88\permil

Ln206 Shouldn't selective water use by plants be included in ET?
**Authors response:** Thanks for pointing this out. We removed 'selective water use by plants' in this sentence.

Ln209 some would argue 7m above the top of the canopy is pretty far out; it is likely outside the subsurface BL near the forest canopy. Since you mention TKE, why don't you show a profile of vertical wind speed and momentum fluxes? It will give you an idea if your sensors are within the canopy subsurface BL.
**Authors response:** We agree that this would be interesting, but we do not have wind profile data available for this cite.

Ln222-223, do you have an explanation of why you found a correlation between dET and dDv but not with d18Ov?
**Authors response:** We think this might be related to the signal to noise ratio, that is better for dD than for d18O. We added this hypothesis to the manuscript.

Ln225-234 these interpretations are based on flawed stats
**Authors response:** Please see our comment above – we changed to multivariate regression.

Ln238. Bowling et al. 2017 is not an appropriate citation here. Remove. **Authors response:** We removed this citation.

Fig 7. Right panel: after leaf coloring - was that diamond or cross symbol?
**Authors response:** We changed the diamonds to crosses.

Ln243-248 & Fig7. Are GMWL and LMWL statistically different? I am skeptical that you can use GMWL and LMWL to contrast impacts by far-field v.s near-field factors.
**Authors response:** The LMWL is 7.4±0.3. Thus, we have a 2-sigma deviation from the GMWL. If we assume a standard distribution of errors, this yields p<0.05. In the revised manuscript we explicitly mentioned the 2-sigma derivation to the interpretation: 'Thus the GMWL with a slope of 8 is at a 2-sigma difference away from the LMWL, yielding a p- value of p<0.05.'

Ln254. entrainment is a diurnal process. Why would you expect entrainment be a factor on seasonal time scales in the first place?
**Authors response:** After reading all the referee reports, we removed the analysis of TKE and u* from our manuscript. This involved also removing this line. However, originally, we wrote this sentence because entrainment integrated throughout the day can be differently strong on different days, this would yield seasonal variability.

---

## Author Comment (AC2) · 19 Feb 2021

**AUTHORS COMMENT: ANSWER TO REFEREE 2**
**'Drivers of the variability of the isotopic composition of water vapor in the surface boundary layer'**

**Referree comments:** black,
**Author comments:** blue
**Changes to the manuscript:** green

In this manuscript, the authors present a dataset of the isotopic composition of water vapor over a forested ecosystem. They combine the measurements of the water vapor with eddy covariance derived estimates of the isotopic ratio of the ET flux. The goal was to test a fundamental hypothesis that the isotopic ratio of water vapor above an actively transpiring surface should respond to the ET flux. Over large scales (i.e. from satellite data) it has certainly been shown that the land surface fluxes of water vapor influence the isotopic ratio of the atmospheric water vapor. The authors conclude that the ET flux has minimal influence on the isotopic ratio of vapor. On diurnal timescales, entrainment drives a midday depletion in the water vapor isotopes that is in opposition to the flux of ET. On seasonal timescales, entrainment rate does not predict the isotopic ratio of the vapor. Rather, it is some combination of processes (ET, rainout and temperature) that collectively influence the isotopic ratio of the vapor.

Firstly, I commend the authors on a very nicely developed dataset and some rather sophisticated analysis of the data. Secondly, I think the question that is posed is interesting and worthwhile particularly to the extent that using water isotopes to trace water fluxes and close hydrological budgets in the atmosphere has a lot of potential in diagnostic analysis of GCMs and transport models. This work contributes to these efforts. However, I found the analysis, on the one hand, to be unnecessarily complex at times (i.e. there were many competing correlations between derivatives) but also overly simple at others (i.e. trying to use a single linear regression model to predict d18Ov). In the end, I think the authors overlooked some simple tests that could have been useful and drew conclusions regarding why d18O/dDv correlated with temperature that are not correct. I would support publication after significant changes are made to the writing and perhaps some additional analyses.

**Authors response:** We thank the anonymous referee for the motivating, detailed and constructive feedback to our manuscript. We understand these general comments on statistical analysis and on the conclusion about the influence of temperature. In the revised manuscript, we added a multivariate regression with fewer variables and drew conclusions based on this. We performed the multivariate regression to reduce the Akaike information criterion (AIC) using a stepwise backward-forward approach. Further, when analyzing simple linear correlations, we focus on fewer variables in particular regarding ET and separated the δD analysis from the δ$^{18}$O analysis in Table 3. However, we still include some correlations, e.g. correlations to IF, to compare with literature data. Concerning the conclusions, in particular about temperature, we tried to be clearer about what we conclude based on the data and suggestions for potential explanations that would need additional measurements. Below we answer the referee's comments in detail.

The authors find that entrainment is the prominent driver of the diurnal cycle in d18Ov except in the morning when transpiration has more of an effect. This finding has been very clearly identified in previous works. See for example: doi.org/10.1002/jgrd.50701 as well as numerous other citations the authors provide. It would seem therefore that the authors should not have been surprised to find this to be true. It would have been surprising, in fact, to find the opposite to be true. I this comment is significant because it affects the entire tone of the paper. The authors should have begun from the perspective that entrainment is the primary driver of diurnal cycles and then sought examples

where the effect of ET emerged.

**Authors response:** Thanks for pointing this out. We added doi.org/10.1002/jgrd.50701 to the discussion of our results:
'This is consistent to the predominant influence of entrainment on the diurnal cycle that has been found by other authors at different field sites (c.f. Lee2007, Griffis2010, Lai2011,Berkelhammer2013).'
We also changed the writing to make it clearer that an influence of entrainment might be expected and our purpose is to QUANTIFY the influence of ET compared to the measured changes in $\delta_v$. e.g. we write:
- 'Our objective is to quantify the influence of local ET on $\delta_v$ in the SBL close to the canopy of a forest ecosystem.'
- '[We] use our direct measurements in combination with PBL heigh t to quantify the influence of ET on $\delta_v$.'
- 'We quantify the influence of local ET on the isotopic composition of the boundary layer by making a quantitative thought experiment. How would local ET influence the delta value of the PBL ($\delta_v$) if local ET would be the only process that (significantly) influences $\delta_v$?. To answer this question, we use isoforcing values, that are based on EC measurements of the magnitude of ET $F_{ET}$ and its isotopic composition $\delta_{ET}$ (see Braden-Behrens2019).'
We hope that this way it is a bit clearer. Please see also the whole section 'Calculation of evapotranspiration-related change in $\delta_v$' of the revised manuscript.

The authors find that the correlation between entrainment rate and the seasonal cycle in d18Ov is weak. They therefore conclude that entrainment is not the critical driver of the seasonal cycle. However, they fail to identify that it is not just how much vapor is entrained but the isotopic ratio of the water vapor that is entrained. With synoptic scale changes in atmospheric circulation the isotopic ratio of water within the free troposphere changes. It would seem quite clear, and maybe I misunderstood this from the manuscript, that it is the isotopic ratio of the free troposphere driven by large scale circulation that drives changes in the midday isotopic ratio above the canopy. Analysis using a lagrangian transport could be deployed (as with many previous isotope studies) to identify how the source of vapor changes and whether it is the source region that explains the seasonal changes.

**Authors response:** We agree, that the isotopic composition of entrained vapor is an important driver and added this to the discussion of seasonal variability:
- 'Concerning entrainment of isotopically lighter air from the free troposphere, the amount of entrained air can vary as well as the isotopic composition of the entrained air which can be studied using lagrangian transport models (see e.g. Aemisegger2014,Pfahl2008).'
We also agree that lagrangian transport analysis would be an appropriate tool to identify changing water sources but this would be beyond the scope of this study. However, we added this to the discussion.
- 'In general, the positive correlation with temperature-related quantities implies that changes in synoptic circulations might be relevant drivers of $\delta_v$. We propose that such changes could be studied by using lagrangian transport models, as have been carried out at other field sites (see e.g. Aemisegger2014, Pfahl2008) and a further analysis of their relation to temperature at the field site might explain parts of the observed variability.'
- 'As the temperature effect is related to the origin and history of air masses (Dansgaard1964, Ambach1968), we propose that lagrangian backtrajectory models would be a useful tool to understand the processes that drive the temperature effect.'

The authors find a strong influence of temperature on d18Ov and call upon a rather confusing role for temperature influencing the fractionation of ET. I find this extremely unlikely. If this was the case, then there should be a very strong relationship between deltaET and temperature. I believe deltaET is more strongly influence by RH or VPD and or LAI.

**Authors response:** Thanks for pointing this out. We agree that the interpretation of the correlation to temperature might have been misleading. In particular because changes due to the variability of $\delta_{ET}$ should be also included in $\frac{d\delta_v}{dt}|_{ET,est}$, the ET-related change in $\delta_v$. Thus, the observed correlation to temperature cannot be explained with changes in T, RH or VPD that yield changes in delta ET. In the revised

manuscript, we removed the misleading interpretation of temperature influencing fractionation.

Revisiting comment #3, changes in synoptic circulation drive both changes in temperature and the d18Ov. The temperature of air masses affect how much rainout has occurred and give rise to a strong relationship between d18Ov and temperature. This is in fact the rationale for why ice core d18O values reflect temperature. I think explaining the relationship between d18Ov and temperature would have benefited from taking a more "first principles" approach and yielding to extensive research already done on this topic.

**Authors response:** Thanks a lot for pointing this out so clearly. We agree that the interpretation of the correlation to temperature might have been misleading, in particular as that synoptic circulation potentially influence both, temperature and delta values. However, we want to point out, that the so called 'temperature effect' is a result of complex processes and our focus was to quantify the influence of local ET, whit we did by analyzing $\frac{\mathrm{d}\delta_v}{\mathrm{d}t}|_{\mathrm{ET,est}}$. We agree, that the (complex) temperature effect is a better explanation for the observed correlation to temperature and include this to our interpretation of the correlation to temperature throughout the manuscript. We changed different parts of the manuscript and now write:

- 'We conclude that the observed seasonal variability of $\delta_v$ is neither dominated by Rayleigh processes, entrainment nor local ET but likely linked to other temperature-related processes such as changes in synoptic circulation.'
- Potential processes that could drive the observed seasonal variability of $\delta_v$ are local ET, cumulative rain-out (Rayleigh distillation) and changes in synoptic circulation.
- At the seasonal time scales the cumulative rainout of an air mass as it ages from its origin (e.g by Rayleigh destillation) is a major driver of the variability of $\delta_v$. This is a complex process that influences $\delta_v$ via the origin of air masses (Ambach1968), the thermodynamic conditions during cooling (see e.g. Dansgaard1964), fractionation during condensation, isotopic exchange between rain drops and the surrounding air and evaporation from rain drops (see e.g. Dansgaard1964). These complex processes yield the 'temperature effect', a positive correlation between condensation temperatures and higher $\delta$-values of precipitation (see e.g. Dansgaard1964).
- A large fraction of 50% of the observed seasonal variability of $\delta_v$ is linked to temperature, indicating a considerable influence of the complex processes that drive the so-called temperature-effect.

The calculation of isoforcing relied heavily on the estimates of PBL height from reanalysis. This concerned me somewhat because there was no good validation of these estimates and it seems the estimates from reanalysis would only be useful if the land cover in the area was homogenous. In other words, is the forested cover of the site representative of the conditions with the reanalysis grid cell?

**Authors response:** We thank the anonymous referee for this comment. This aspect was indeed missing from the manuscript, so we have now mentioned the landcover and the representativity of the grid cell for the study site in the revised text. Indeed, while the tower itself is situated in a forest, this forest patch is only a couple km wide, and the landscape at a larger scale is a mixture of such forest areas and agriculture. The spatial scale of the forest is such that the boundary layer formation will be routinely driven not only by the forest patch but by the larger mixed landscape. Comparing the land cover and the relative fractions of forest and agriculture for the ERA5 reference gridcell relative to the relevant area around the study, we find that the grid cell is representative for the study site. We have added this point to the revised text:

'The land cover within this grid cell is characterized by a mixture of forested patches and agricultural land, with a relative contribution of about 50% each. A similar mixture of forest and agriculture is found at the study site at a spatial scale relevant for the formation and growth of the PBL. Due to the proximity and the comparable land cover of the entire grid cell and the area surrounding the tower at the measurement site, we consider the boundary layer height estimate of the grid cell to be representative of conditions at the study site.'

The authors discuss error estimates of PBL height but it was not clear how these error values were assimilated in the analysis.

**Authors response:** Unfortunately, we do not have error estimates for all datapoints, but only for a subset of datapoints. Thus, we show the discussion of uncertainties and not propagate those uncertainties further. In the revised manuscript, we added more information about the PBL-heigt data source and its uncertainties for the uncertainty or delta_ET we refer to the previously published manuscript on https://doi.org/10.1016/j.agrformet.2019.01.035, that analyse the uncertainty of delta_ET as a function of the magnitude of ET.

Secondly, the authors note that their assumption that the isotopic ratio of water vapor is well mixed is likely incorrect. This has been shown by other studies using gradient and flux gradient approaches. What are the effects of this assumption on the isoforcing estimate? What if the authors assumed a gradient with log form up the top of the PBL using previous studies? My point is that if the authors know this assumption is incorrect it would be valuable to assess the impact of this assumption on their analysis using a sensitivity approach.

**Authors response:** This is a very good idea, but we do not have the necessary data/boundary conditions to perform such an analysis.

I was surprised come to the end of the paper and never see a figure or actual discussion on the estimates of delta ET. The estimates of delta ET were assimilated into numerous analyses but, after all, if the study is looking at how delta ET affects delta V, the readers should see delta ET. The authors need to present this data and analyse it directly before using it in more sophisticated approaches. How does delta ET vary through the season? Was it affected by soil moisture and VPD that might change T and E partitioning? Did delta ET relate to total ET rates or greeness/LAI? Does it change after rainfall events? An analysis of the drivers of delta ET are a necessary complement to the other analyses presented.

**Authors response:** Our measurements of delta_ET is presented and discussed in a previous paper https://doi.org/10.1016/j.agrformet.2019.01.035. This manuscript focusses on using this data to better understand $\delta_v$, so we do not want to present the same data twice. However, we agree that changes in delta_ET are important, thus we added a brief summary of the results concerning delta_ET of our previous paper to this manuscript and refer to it for further details.

- [… ] the diurnal cycle of $\delta_{ER}$ does not dominate the diurnal cycle of IF. These diurnal cycles are shown in a previous manuscript (see Braden-Behrens2019). In brief, $\delta_{ET}$ rose throughout the day, indicating non-steady-state conditions both $\delta$ values and over all seasons, except for $\delta D_{ET}$ in summer.
- In brief, $\delta_{ET}$ spanned a range of -19 to 0 permil for $\delta 18O_{ET}$ and of -140 to -25 permil for $\delta D_{ET}$. with a complex seasonal shape and larger uncertainties for smaller ET, (see Braden-Behrens2019 for details).

Small comments:

When a variable is introduce the correct grammar (I think) is like this: : : "Temperature, T, is related to latitude." Or "Temperature (T) is related to latitude."

**Authors response:** Thanks for pointing this out. We try to change this throughout the manuscript.

28: Unclear why sublimation of snow was listed under "precipitation removal" processes. This would be a surface flux process.

**Authors response:** We removed it.

54: The R2 value between C and d18O/dD were just listed in the previous paragraph so this sentence felt redundant.

**Authors response:** This sentence refers to the study of Griffis 2016 does not use R^2 values, but a different method. We changed this sentence to be more clear: 'In particular, even at a height of 185 m above a crop/grassland canopy, (Griffis2016) estimate the relative contribution of ET to range from 0 to close to 100%,

with a median of 34% based on a two-end-member isotope mixing model (Griffis2016)'

60: Lots of other studies over forests not considered here: Continuous measurements of atmospheric water vapour isotopes inwestern Siberia (Kourovka) Stable Water Isotopes Reveal Effects of Intermediate Disturbance and Canopy Structure on Forest Water Cycling Response of water vapour D-excess to land–atmosphere interactions in a semi-arid environment I would say broadly that the literature available on this topic was under-cited.
**Authors response:** We added these studies to the discussion, in particular to Table 2.

145-155: This extended quotation from ERA5 manual is not appropriate. The authors should explain the process of error estimation in their own words.
**Authors response:** This extended quotation from ERA5 manual has been replaced in a complete reworking of this section on PBL height.

As noted above, it is also unclear how this error was assimilated in the analyses that follow.
**Authors response:** Please see our comment above.

163: Missing "space" before the sentence begins.
**Authors response:** We changed this.

s "site" not "cite"
**Authors response:** We changed this.

171: "However" is the wrong word here because this sentence does not contradict the previous one it supports it.
**Authors response:** We removed this.

176: The comma should be after "h" not after "both"
**Authors response:** We changed this.

178: if the nighttime data is not meaningful, I would recommend excluding it. As you note, when the value approaches 0, the equation becomes very unstable.
**Authors response:** Yes, we excluded it from the analysis. We added a sentence about this to the manuscript.

181: When you write ddv/dt is this dt_iso or dt_meas. Truthfully, I found the comparisons between the many derivatives quite hard to follow and perhaps not the most useful way to analyze the dataset.
**Authors response:** In the revised manuscript, we changed the notation to be more clear. We also added a more detailed description about the purpose and the underliing assumptions of the ET-related estimate $\frac{d\delta_v}{dt}|_{ET,est}$.
We now only use $\frac{d\delta_v}{dt}|_{ET,est}$ for the estimated quantity and $\frac{d\delta_v}{dt}|_{meas}$ for the measured quantity to avoid this confusion. Additionally, we changed the table that presents the different regressions to be more clear (using less parameters) and changed the analysis to a multilinear regression: We performed the multivariate regression to reduce the Akaike information criterion (AIC) using a stepwise backward-forward approach.

Figure 1: Standard error should be reported around composite diurnal cycles.
**Authors response:** Unfortunately, we do not have uncertainty estimates for all datapoints and quantities.

193: "being"
**Authors response:** We changed this.

206: I was confused as to what the authors mean by Rayleigh distillation in this context. Is this condensation onto the surface such as through dew or is this the collective rainout of the air mass as it ages from its origin?

**Authors response:** We changed this sentence to: 'Potential processes that could drive the observed seasonal variability of $\delta_v$ are local ET, cumulative rain-out (Rayleigh distillation) and changes in synoptic circulation.'

206: Also, because all of these processes are important to the hydrological balance, it would seem that linear univariate models are not really appropriate or useful. Perhaps multivariate non-linear models would be better suited for partitioning the relative controls.

**Authors response:** Yes, we changed the statistical analysis a multivariate regression to reduce the Akaike information criterion (AIC) using a stepwise backward-forward approach. We are aware, that by this we cannot fully model the isotopic composition, but our goal is only to identify potential drivers. For this purpose, we decide for a linear model instead of a nonlinear model, because we want to avoid overfitting.

207: "between"
**Authors response:** We changed this.

208: missing closed parenthesis at end of paragraph.
**Authors response:** We changed this.

209: Earlier you discuss the inlet being 10 m above canopy but here you say 7 m. Not a big deal but better to just be consistent.
**Authors response:** We changed this to 7m throughout the manuscript.

Figure 6 and associated discussion on Rayleigh Distillation: The assumption that a single distillation model (i.e. a linear fit to d18O vs. log(C)) assumes that a common source but experiencing different degrees of rainout. This is not true. So you could really have multiple plausible distillation models that would give rise to "messier" scatter plot of your data.
**Authors response:** Yes, we added this to the discussion:
-   'However, it is worth to point out, that this discussion of Rayleigh destillation is based on the assupmtion of one single destillation model. Thus, some of the additional variability in the relationship between $\delta_v$ and $\log(C')$ in Fig. 6 might also be explained by multiple distillation processes.'

253: How does delta ET relate to precipitation? This could give you some insight into the fractionation of ET relative to the source. Does it change during the year?
**Authors response:** Please see our comment above: We added a brief description of the seasonality (and uncertainty) of delta_ET to our manuscript.

261-262: The authors write: "In general, the correlation between temperature and v might be linked to temperature dependent fractionation at the sites of evaporation." What are the sites of evaporation being referred to here? Local ET? Nearby lakes that might supply atmosphere? The ocean source?
**Authors response:** After reading your general comments about synoptic changes, we removed this part of the discussion, because we agree that this is unlikely. We removed this misleading interpretation from the manuscript and thank the anonymus referee for pointing this out.

---

## Author Comment (AC3) · 19 Feb 2021

**AUTHORS COMMENT: ANSWER TO REFEREE 3**
**'Drivers of the variability of the isotopic composition of water vapor in the surface boundary layer'**

Referree comments: black,
Authors response: blue
Changes to the manuscript: green

This manuscript deserves final publication in BG after a few minor corrections and editorial adjustments. The authors present their analysis in a clear and logical fashion. The dataset was obtained with a well-tested instrumental system. A key strength of this analysis is the measurement of the vapor isotopic flux to inform interpretation of physical drivers of the observed vapor isotopic variability.

**Authors response:** We thank the anonymous referee for the motivating, positive and constructive feedback to our manuscript, below we answer the referee's comments in detail.

Line 150: Unlike other independent variables, here the PBH height is model-derived. Can you comment on efforts (by you or others) to evaluate the ERA h against observed h for your geographic region?

**Authors response:** Thanks for pointing this out. As a reanalysis product, PBLH is model-derived, however not without significant measurement data assimilation. We have added a description of a comparative study (Seidel et al. 2012) including our geographic region, assessing PBL height from reanalysis relative to radiosonde measurements. This study assesses the same Bulk-Richardson approach for estimation of PBLH, which is used in ERA5. We have now included some of their most important findings in the revised text. We had also worked on comparing ERA5 with alternative estimates of PBLH, such as the work by McNaughton and Springgs (1986) using data from Cabauw, however the currently available ERA5 data do not extend back to the same time. We added the following to the manuscript: 'Regarding the uncertainty of the boundary layer height of the ERA5 reanalysis product relative to radiosonde observations, (Seidel2012) found good agreement between the two measures with a relative uncertainty of ERA5 of generally less than 20% for sufficiently deep boundary layers with a height of more than 1 km, and up to 50% uncertainty for more shallow boundary layers. ERA5 estimates of h tended to be larger than radiosonde measurements due to the difficulty of accurately modelling h under stable conditions.'

Line 190: The message here is quite clear. Can you comment on the implication for Keeling mixing line analysis?

**Authors response:** Thanks a lot for this suggestion, we added the following sentence to our manuscript: 'Based on this observation, we conclude that measuring the isotopic composition of ET with Keeling mixing line applications would not be justified at our measurement site, because these applications assume a single source mixing with background air. When entrainment is occurring as in our case, we have two sources thus violating the Keeling plot assumption resulting in a bias. In a Large eddy simulation study, it was shown that such biases are particularly pronounced for water vapor isotopes reaching several permille (Lee et al. 2012).'

Line 207: typo "betreen"
**Authors response:** Thanks for finding this. We changed it.

Line 215: you mean ": : : when we expect NO transpiration: : :"?
**Authors response:** We expect transpiration through the leaves in the period 'green leaves'. Maybe this sentence is a bit long, so we rephrased it to be more clear.
'[...]we expect transpiration when green leaves are present. The period with green leaves (refered to in Fig. 5 started with leaf unfolding on 19. April and lasted until leaf senescence on 6. October 2016. During this period, there are significant ($p<10^{-5}$) but weak ( $R^2$ app. 0.17 )

correlations between $\delta_v$ and the corresponding isoforcing values IF as well as the isoforcing related change [...].

Line 233: Some people consider the lack of correlation between vapor delta and concentration as indicative of Rayleigh distillation associated with atmospheric convection.
(When an air parcel movement span a large vertical distance, condensation occurs over a large range of temperature.)
**Authors response:** This is an interesting point. We added the following discussion of deviations from Rayleigh distillation to our manuscript:
'This deviation from Rayleigh distillation in summer might be related to other relevant fluxes. Further, derivations from Rayleigh distillation have also been found as a result of deep convection (Tharamal2017) based on modelling. However, it is worth to point out, that this discussion of Rayleigh distillation is based on the assupmtion of one single destillation model. Thus, some of the additional variability in the relationship between $\delta_v$ and $\log(C_{H2O})$ in Fig. 6 might also be explained by multiple distillation processes.'

Figure 1: ET unit is incorrect. The unit carried by IF is different from that shown in Figure 4
**Authors response:** Thanks for pointing this out, this was a typo. We changed the units accordingly.

Figure 7 left panel: I don't see rain data
**Authors response:** We removed 'rain data' in the legend. We do not show it here, because we focus on the LMWL which is based on rain data.

Figure 7 right panels: These basically reveal seasonal pattern of vapor d-excess. Can you comment on diurnal pattern of vapor d-excess and its implications?
**Authors response:** Thanks for this suggestion, we plotted the diurnal cycle of d-excess, but we found it not straightforward to interpret and thought the discussion would be beyond the scope of our manuscript. Thus decided to not show it in the revised manuscript. However: Here we show the diurnal cycle of local d excess:

[Figure]

Figure 2 & Table 3: How did you obtain TKE?
**Authors response:** This was based on our EC measurements. However, in the review process, we decided zo remove TKE from the analysis because of ... (cf. Referee 1)